# A glycan receptor kinase facilitates intracellular accommodation of arbuscular mycorrhiza and symbiotic rhizobia in the legume *Lotus japonicus*

**Simon Kelly**[1], **Simon B. Hansen**[1], **Henriette Rübsam**[1], **Pia Saake**[2], **Emil B. Pedersen**[1], **Kira Gysel**[1], **Eva Madland**[1], **Shunliang Wu**[3], **Stephan Wawra**[2], **Dugald Reid**[1], **John T. Sullivan**[4], **Zuzana Blahovska**[1], **Maria Vinther**[1], **Artur Muszynski**[5], **Parastoo Azadi**[5], **Mikkel B. Thygesen**[3], **Finn L. Aachmann**[6], **Clive W. Ronson**[4], **Alga Zuccaro**[2], **Kasper R. Andersen**[1], **Simona Radutoiu**[1], **Jens Stougaard**[1]*

1 Department of Molecular Biology and Genetics, Aarhus University, Aarhus, Denmark, 2 Cluster of Excellence on Plant Sciences (CEPLAS), Institute of Plant Sciences, Cologne, Germany, 3 Department of Chemistry, University of Copenhagen, Frederiksberg, Denmark, 4 Department of Microbiology and Immunology, University of Otago, Dunedin, New Zealand, 5 Complex Carbohydrate Research Center, University of Georgia, Athens, Georgia, United States of America, 6 NOBIPOL (Norwegian Biopolymer Laboratory), Department of Biotechnology and Food Science, NTNU Norwegian University of Science and Technology, Trondheim, Norway

☯ These authors contributed equally to this work.
* stougaard@mbg.au.dk

**Data Availability Statement:** All relevant data are within the paper, its Supporting information files

## Abstract

Receptors that distinguish the multitude of microbes surrounding plants in the environment enable dynamic responses to the biotic and abiotic conditions encountered. In this study, we identify and characterise a glycan receptor kinase, EPR3a, closely related to the exopolysaccharide receptor EPR3. *Epr3a* is up-regulated in roots colonised by arbuscular mycorrhizal (AM) fungi and is able to bind glucans with a branching pattern characteristic of surface-exposed fungal glucans. Expression studies with cellular resolution show localised activation of the *Epr3a* promoter in cortical root cells containing arbuscules. Fungal infection and intracellular arbuscule formation are reduced in *epr3a* mutants. *In vitro*, the EPR3a ectodomain binds cell wall glucans in affinity gel electrophoresis assays. In microscale thermophoresis (MST) assays, rhizobial exopolysaccharide binding is detected with affinities comparable to those observed for EPR3, and both EPR3a and EPR3 bind a well-defined β-1,3/β-1,6 decasaccharide derived from exopolysaccharides of endophytic and pathogenic fungi. Both EPR3a and EPR3 function in the intracellular accommodation of microbes. However, contrasting expression patterns and divergent ligand affinities result in distinct functions in AM colonisation and rhizobial infection in *Lotus japonicus*. The presence of *Epr3a* and *Epr3* genes in both eudicot and monocot plant genomes suggest a conserved function of these receptor kinases in glycan perception.

and the NCBI BioProject accession PRJNA953045 (http://www.ncbi.nlm.nih.gov/bioproject/953045).

**Funding:** SK, SBH, HR, KG, EM, SW, DR, ZB, MV, MBT, KRA, SR and JS is supported by the research project Engineering Nitrogen Symbiosis for Africa (ENSA), which is funded through a grant to the University of Cambridge by the Bill & Melinda Gates Foundation (OPP11772165). JS acknowledge support from the European Research Council (ERC) under the European Union's Horizon 2020 research programme (grant agreement No. 834221). KRA and HR acknowledge support from the Danish Council for Independent Research (9040-00175B). AM and PA acknowledge support from the U.S. Department of Energy, Office of Science, Basic Energy Sciences, Chemical Sciences, Geosciences and Biosciences Division, under award #DE-SC0015662. AZ, PS and SW acknowledge support from the Cluster of Excellence on Plant Sciences (CEPLAS) funded by the Deutsche Forschungsgemeinschaft (DFG, German Research Foundation) under Germany's Excellence Strategy–EXC 2048/1–Project ID: 390686111. FLA acknowledge the Research Council of Norway who has contributed though the grant 226244 (Norwegian NMR platform- NNP). The funders had no role in study design, data collection and analysis, decision to publish, or preparation of the manuscript.

**Competing interests:** The authors have declared that no competing interests exist.

**Abbreviations:** AM, arbuscular mycorrhiza; EPS, exopolysaccharides; IT, infection thread; MST, microscale thermophoresis; pLDDT, predicted local distance difference test; PMAA, partially methylated alditol acetate; RNS, root nodule symbiosis; ROS, reactive oxygen species.

## Introduction

Plant symbiosis with arbuscular mycorrhizal (AM) fungi of the Glomeromycota is found in 80% to 90% of all land plants. Fossil records from early land plants and the presence of AM symbiosis in liverworts, hornworts, lycophytes, and ferns suggest that AM symbiosis may have evolved in the earliest land-colonising plants [1,2]. Conservation of symbiosis genes in algae further suggests that a common genetic programme governing AM symbiosis predated and has been maintained in land plants [3]. Mutant studies in legumes and non-legumes have identified part of this genetic program. The "common symbiosis pathway" shared with plant–rhizobial symbiosis is required for normal mycorrhizal invasion and root colonisation [4,5]. Prior to AM colonisation of roots, pre-symbiotic signalling establishes the communication process to distinguish AM fungi from pathogens and other soil fungi. Germination of AM spores is induced by strigolactones secreted by plant roots [6,7]. The exact nature of the reciprocal fungal signal(s) found in germinating spore extracts is less well defined. Both chitin from the fungal cell wall and lipochito-oligosaccharides (MYC-factors) have been implicated [8,9]. Recent results in *Medicago truncatula* (*Medicago*) *nfp cerk1* double receptor mutants impaired in both lipochito-oligosaccharide and chitin perception show that a combination of fungal chitin and lipochito-oligosaccharides triggers the plant signal transduction through the common symbiosis pathway [10]. Other compounds may also contribute. In rice, a butanolide signal perceived by the D14L α/β-fold hydrolase receptor is essential for AM infection, but at this point, the origin of the butanolide signal is unknown [11].

Following activation of the plant cellular program(s) AM hyphae penetrate the root epidermal and outer cortical cells via a pre-penetration apparatus resulting from cellular rearrangements in a process involving symbiotic genes [12]. Arbuscules are formed by intracellular invasion of the inner cortical cells at a position where the plasma membrane invaginates and a subtending pre-penetration apparatus is established. Following entry at a single position, forming a trunk, the fungal hyphae branch out into a finely branched structure surrounded by a peri-arbuscular membrane derived from the plant plasma membrane. In the legume plants analysed so far, formation of these feeding and nutrient-exchange arbuscule structures occurs mainly in the inner cortical cells [13,14]. Mutant studies have shown that a CCaMK-CYCLOPS-DELLA complex together with the RAM1 transcription factor are required for arbuscule formation [15–18]. However, the molecular mechanism controlling cell preference and the signal exchange that directs infecting AM towards the inner cortical cells forming intracellular arbuscules remain unidentified.

In *Lotus*, nitrogen-fixing symbiosis with *Mesorhizobium loti* (*M. loti*) involves a two-step recognition process for intracellular infection that operates both at the epidermis and in the cortical tissue. In this compatibility surveillance mechanism, bacterial exopolysaccharides (EPS) are perceived by the EPR3 exopolysaccharide receptor [19]. This perception is downstream of primary lipochito-oligosaccharide (Nod factor) signalling. *M. loti exoU* mutants that produce truncated forms of EPS are severely impaired in intracellular infection thread (IT) formation and consequently in the formation of nitrogen-fixing nodules [20]. Characterisation of *Lotus* mutants that restore formation of nitrogen-fixing nodules following inoculation with the *exoU* mutant and *in vitro* binding assays identified EPR3 as a receptor for EPS [19,21]. The structure of the EPR3 ectodomain was recently resolved, revealing EPR3 to be the founding representative of a unique class of plant RLKs with a distinctive modular ectodomain arrangement [22]. *In vitro*, the EPR3 ectodomain binds EPS of several rhizobial species, including micro-symbionts unable to nodulate *Lotus*, suggesting that the receptor may have a broader role in monitoring glycans from various root-associated microbes [22].

In this study, we report on the identification and characterisation of a novel glycan receptor kinase in *Lotus* that we have designated EPR3a. We have characterised the symbiotic phenotypes of *epr3a*, *epr3*, and *epr3a epr3* mutants inoculated with AM, wild-type *M. loti* and *M. loti exoU* EPS mutants. Mutant phenotypes and biochemical characterisation suggest that the EPR3a and EPR3 receptors most likely signal independently with overlapping downstream pathways during rhizobial infection. Distinct differences in AM symbiosis were observed where expression of *Epr3a* and an associated mutant phenotype were found.

## Results

### *Epr3a* induction in arbusculated cells

The *Epr3a* gene was identified as (LotjaGi4g1v0157000) and located on chromosome 4 of a recently released *de novo* genome assembly covering 554 Mb of the *Lotus japonicus* Gifu accession [23]. Protein alignment against the previously identified members of the LysM-RLK family in *Lotus* revealed that EPR3a is most closely related to the EPS receptor EPR3 with an overall 64% amino acid identity (Fig 1A and S1 Fig) [19]. A tertiary structure similar to EPR3, with three putative ligand-binding modules in the extracellular domain, a transmembrane domain, and a kinase domain, is predicted for EPR3a. The recently resolved crystal structure of the EPR3 ectodomain revealed characteristic M1, M2, and M3 ligand-binding modules in the extracellular domain [22]. Prediction modelling of the EPR3a ectodomain using Alphafold2 [24,25], indicates that EPR3a and EPR3 ectodomains resemble each other (Fig 1B). Importantly, the distinctive ectodomain structure of a βαββ M1 conformation, a βαβ M2 conformation, and a classical LysM βααβ secondary structure of M3 is conserved between these two receptors (Fig 1B and S2 Fig). A wider search in the genome databases shows that this class of LysM-RLKs is widely conserved in the plant kingdom [22], with most plants able to form mycorrhizal and/or bacterial endosymbiosis encoding at least one EPR3-type receptor (S3 Fig) [26]. Interestingly, functional Epr3 and Epr3a genes were not found in genomes of *Parasponia* and *Aeschynomene evenia* plants [26,27], both of which have endosymbiosis with mycorrhiza and rhizobia.

To explore and compare the functional role of EPR3a and EPR3, we investigated whether EPR3a could be involved in AM symbiosis. The expression of *Epr3a* and *Epr3* was examined through qRT-PCR analysis in a time series following inoculation of *Lotus* roots with AM spores. *Epr3a* is induced only in roots during AM symbiosis, with an expression pattern mirroring that of the *Pt4* phosphate transporter, an AM symbiotic marker associated with arbuscule formation [28] (Fig 2A and S4A Fig). In contrast, *Epr3* expression did not differ between mock and AM treatments except for a minor transient induction at 2 dpi. To determine the cellular expression, promoter activity during AM symbiosis was examined in transgenic *Lotus* roots using p*Epr3a*:GUS or p*Epr3*:GUS reporter constructs. Complementary to the qRT-PCR result, *Epr3a* promoter activity was observed in response to AM spore inoculation, while no *Epr3* promoter activity was detected (S4B Fig). In order to determine the spatiotemporal regulation of *Epr3a* promoter activity in AM-colonised roots, histochemical staining and microscopy were performed on transgenic roots expressing the promoter-GUS reporter constructs. *Epr3a* expression was found to be specifically induced in inner cortical cells associated with intracellular arbuscule formation (Fig 2B).

### Mycorrhization phenotype of *epr3a* mutants

The *Epr3a* expression pattern suggested that EPR3a might be involved in arbuscule formation during AM symbiosis. To examine the potential functional roles of *Epr3a*, two independent LORE1 mutant alleles, *epr3a-1* and *epr3a-2*, were isolated from the *Lotus* LORE1 mutant

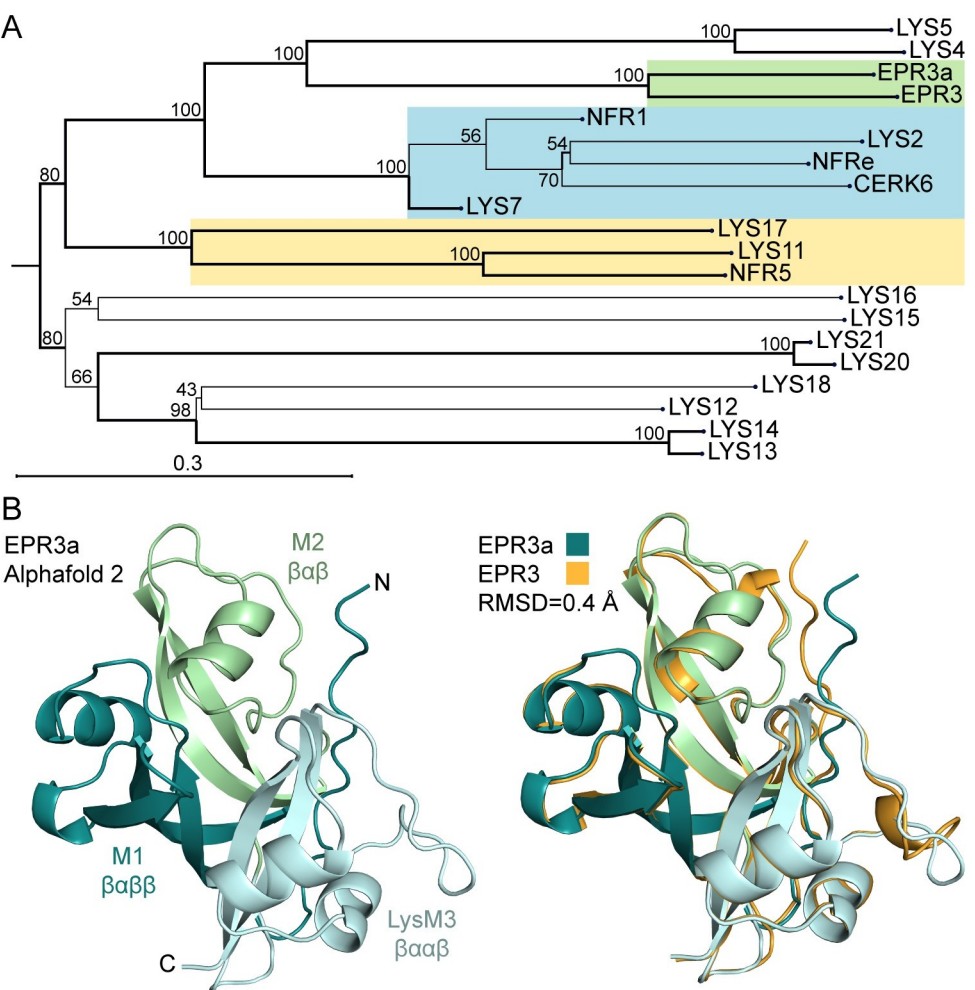

**Fig 1. *L. japonicus* LysM-RLK family and EPR3a ectodomain structure.** **(A)** Phylogenetic tree of the *L. japonicus* Gifu LysM-RLK protein family. EPR3-type (green), NFR1-type (blue), and NFR5-type (yellow) members are highlighted. Bootstrap values are indicated, and branches with values >80 are in bold. **(B)** EPR3a ectodomain Alphafold2 prediction model [24,25]. EPR3a structurally resembles the EPR3 ectodomain crystal structure with a Cα superposition RMSD of 0.4 Å [22]. Like EPR3, EPR3a contains two non-canonical M1 (βαββ) and M2 (βαβ) domains in addition to a conventional LysM3 (βααβ). See S1 Data and S1 Information for underlying data.

resource established in the *L. japonicus* Gifu accession [29] (S5 Fig and S1 Table). An *epr3a epr3* double mutant was isolated from crosses of *epr3-11* [19] and *epr3a-2* mutants. After spore inoculation, *epr3a* single mutants and the *epr3a epr3* double mutants showed a comparable significant reduction in arbuscule formation, together with an increase in the presence of AM vesicles (Fig 3). No difference in AM symbiosis phenotype was observed for the *epr3* mutant compared to Gifu (Fig 3). Arbuscule structure in *epr3a* mutants appears indistinguishable from those in wild-type plants, suggesting a role for EPR3a in fungal entry into cortical cells rather than in arbuscule development.

## EPR3a binds branched glycans

Localisation of *Epr3a* expression in arbusculated cells and the reduced frequency of arbuscule development in *epr3a-1* and *epr3a-2* mutants suggest that EPR3a could be involved in

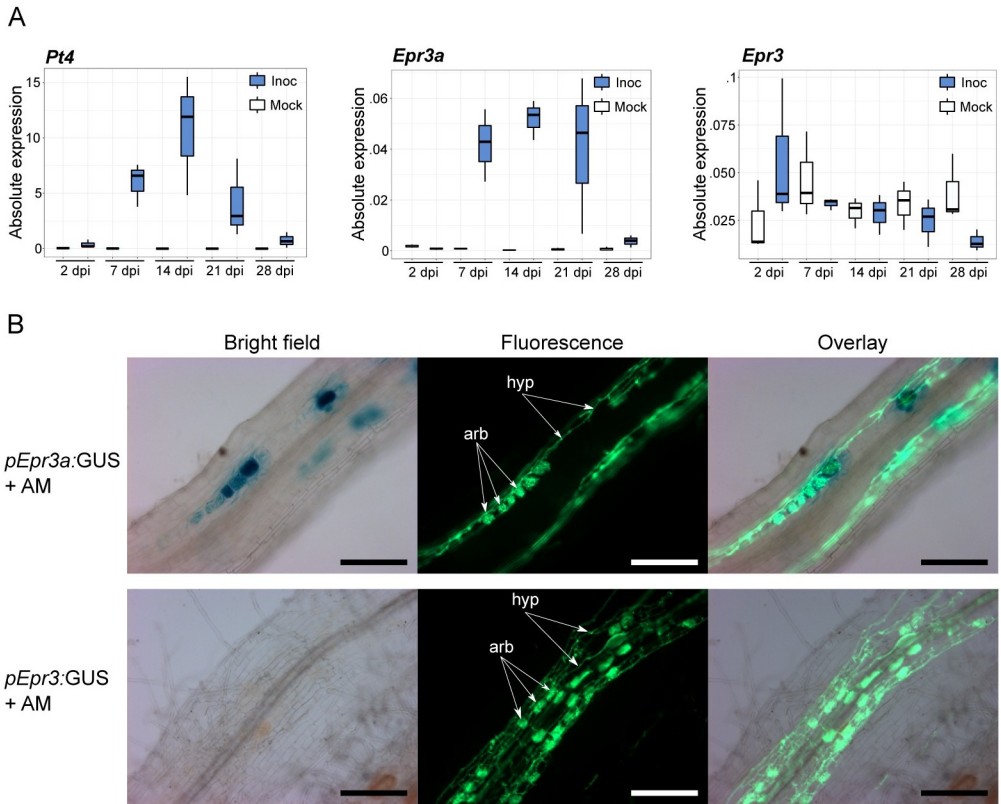

**Fig 2. *Epr3* and *Epr3a* expression in response to AM. (A)** qRT-PCR analysis of *Pt4*, *Epr3a*, and *Epr3* expression in Gifu during the establishment of AM symbiosis. Results are from 3 biological replicates. **(B)** Transgenic roots expressing *pEpr3a*:GUS or *pEpr3*:GUS were inoculated with AM spores. Concomitant visualisation of GUS activity (blue staining) and fungal structures (green fluorescence) 6 wpi demonstrates *Epr3a* promoter activity is localised to cortical cells harbouring intracellular mycorrhiza. Examples of AM arbuscules (arb) and hyphae (hyp) are indicated; scale bars = 200 μM. See S2 Information for underlying data. AM, arbuscular mycorrhiza.

communication, perceiving a secreted signal or a surface exposed molecular pattern. To examine the ability of EPR3a to perceive ligands the ectodomain of EPR3a was expressed in insect cells and purified (S6A Fig). Independent purifications of EPR3a ectodomain showing similar thermostability were used in biochemical assays (S6B Fig).

Fungal cell walls consist primarily of chitin polymers, β-1,3 glucan and β-1,6 branched β-1,3 glucan polymers. β-1,3 glucan was previously localised to the AM cell wall by immunogold labelling [30]. Here, we show the presence of β-1,6 glucan in the extracellular matrix (EPS) and cell wall of AM using the fluorescently labelled β-glucan-binding *Si*FGB1 lectin specific for β-1,6 glucan (Fig 4) [31]. Purified glucans from AM fungal cell walls or EPS are not available; therefore, the binding capacity of EPR3a ectodomains was tested in native affinity gel electrophoresis using laminarin, which contains a mix of β-1,6 branched β-1,3 glucans also found in fungal cell walls (see NMR characterisation below). Homogenised unbranched polymeric shrimp shell chitin was used as chitin polymer. A clear concentration-dependent retention of the EPR3a ectodomain was detected using affinity gel electrophoresis with laminarin from *Eisenia bicyclis*, while no retention was observed with laminarin from *Laminaria digitata*, scleroglucan, pustulan, or chitin (Fig 5A and S7A Fig). The opposite behaviour was found for the ectodomain of the *Arabidopsis thaliana* chitin receptor CERK1, which was retained by

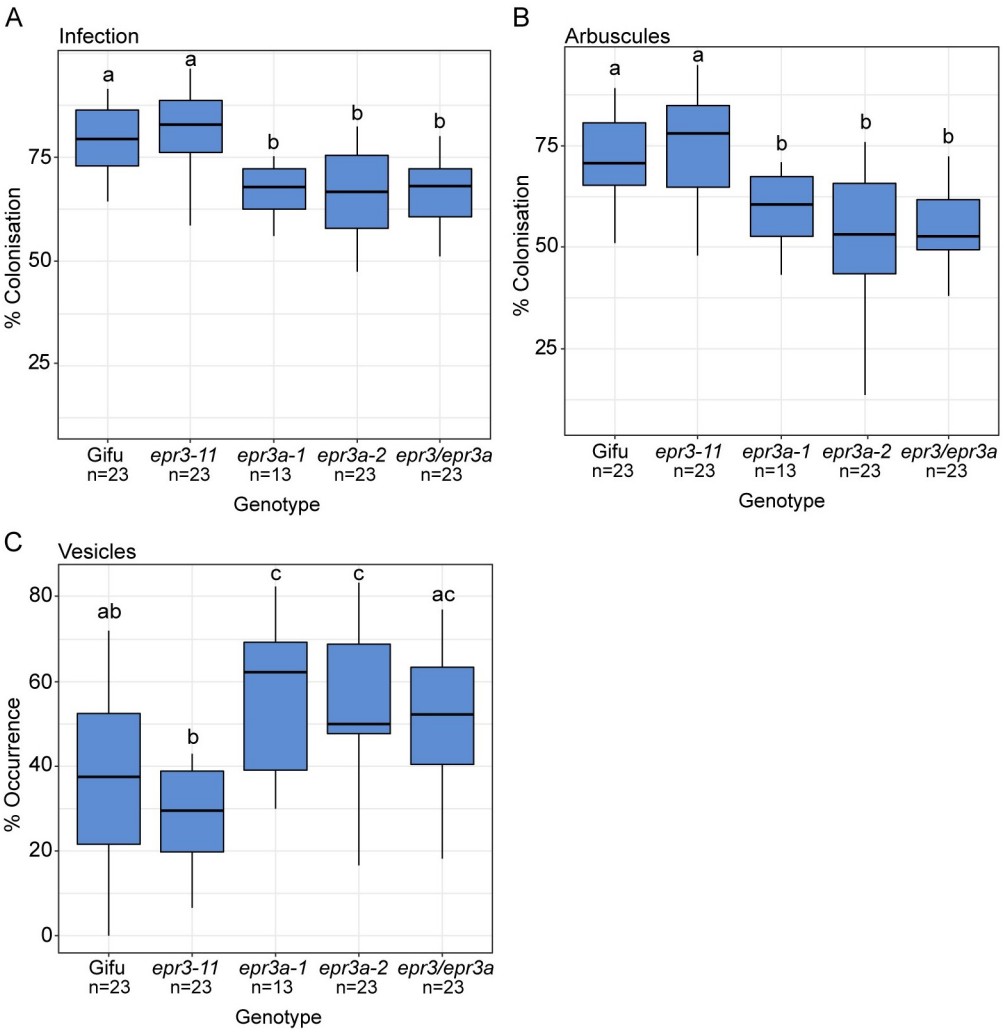

**Fig 3. Arbuscular mycorrhiza phenotyping of the indicated genotypes 6 wpi with AM.** The colonisation rates of **(A)** fungal infection, **(B)** arbuscule formation, and **(C)** vesicle occurrence were scored. Statistical comparisons between genotypes for each of the infection events are shown using ANOVA and Tukey post hoc testing with *p* values (<0.05), as indicated by different letters. See S1 Data for underlying data. AM, arbuscular mycorrhizal.

chitin but not by laminarin. Migration of the EPR3 ectodomain was not influenced by β-glucan or chitin (Fig 5A and S7A Fig).

Retention by *E. bicyclis* laminarin suggests that the β-1,6 branches are likely recognised in the context of β-1,3 polymer cores. Glucosidic linkage analysis of the *L. digitata* and *E. bicyclis* laminarins confirmed the higher degree of β-1,6 branches as well as extended β-1,6 motifs of *E. bicyclis* laminarin (S8A and S8B Fig). In addition, the molecular size of the two laminarins differed substantially. Analytical size exclusion chromatography (SEC) analysis estimates the *L. digitata* laminarin size to be ≈ 5 kDa, while *E. bicyclis* was around 9-fold larger, with an average size of 44 kDa (S8C Fig). Molecular weights and β-1,3/β-1,6 ratios determined for both laminarins are consistent with previous reports [32,33]. Estimates of oligomerisation degrees translate respectively to 30-repeat and 270-repeat oligomers, likely not insignificant considering EPR3a retention capabilities. Estimating molar concentrations of laminarins in the assays based on laminarin molecular weights gives a concentration of 1 mM for *L. digitata* gels, while

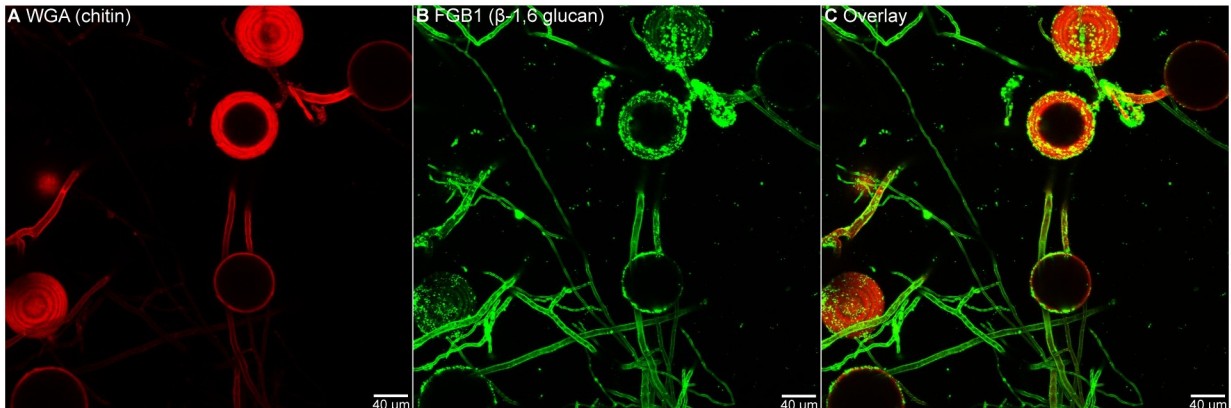

**Fig 4. AM fungi produce an extracellular matrix rich in β-glucans. (A)** Chitin was visualised using fluorescently labelled WGA. Chitin was found in the inner fungal wall of spores and large hyphae. **(B)** β-1,6 glucan was visualised using fluorescently labelled fungal glucan-binding 1 (FGB1). β-1,6 glucan was found as part of the outer cell wall or in loosely associated extracellular matrix (EPS) surrounding spores, large hyphae as well as thin hyphae. **(C)** Overlay of chitin and β-glucan visualisation, confirming that β-glucan is abundant relative to chitin in the EPS of AM fungi. AM, arbuscular mycorrhiza; EPS, exopolysaccharide; WGA, wheat germ agglutinin.

*E. bicyclis* gels contained down to 60 μM of the β-glucan. This further illustrates the selectivity of EPR3a in recognising a specific β-1,3/β-1,6 branched glucan structure of the *E. bicyclis* laminarin.

Microscale thermophoresis (MST) binding assays were performed to further characterise EPR3a β-glucan binding. EPR3a showed higher affinity for binding *E. bicyclis* laminarin compared to laminarin from *L. digitata*, with a $K_d \approx 250$ μM for *E. bicyclis* and >3 mM for *L. digitata* (Fig 5B and S7B Fig). NMR assignment of *E. bicyclis* and *L. digitata* laminarin showed that both samples contain a mixture of several β-1,3/β-1,6 branched carbohydrate species, making it challenging to establish the exact molecular pattern that EPR3a binds. The structures differ in the 1,6 branches, where *L. digitata* laminarin has terminal glucosyl units in low abundance, while *E. bicyclis* laminarin has more complex patterns in higher abundance (S9 Fig). A well-defined β-1,3/β-1,6 decasaccharide, derived from the EPS of the endophytic fungus *Serendipita indica* or the pathogenic fungus *Bipolaris sorokiniana*, was recently shown to scavenge reactive oxygen species (ROS) and to positively regulate fungal colonisation in barley [34]. EPR3a binds the *S. indica* β-1,3/β-1,6 decasaccharide in two events, with the first binding event having an estimated $K_d = 51 \pm 23$ μM, indicating binding specificity for a defined β-1,3/β-1,6 repeating pattern (Fig 5C and 5E). The low-affinity binding of *L. digitata* laminarin and the secondary binding for the *S. indica* decasaccharide ($K_d$ in mM range) are possibly artificial due to changes in solution viscosity (gelation), which is a well-known property of β-glucans at high concentrations. To assess the binding specificity for β-1,3-glucans in the absence of β-1,6 branches, the affinity for two linear β-1,3 oligosaccharides, laminarihexaose and laminaripentaose, was assayed. EPR3a bind both β-1,3 oligosaccharides with low affinity (in the range of 600 to 800 μM), corresponding to at least a 10-fold lower affinity compared to the *S. indica* β-1,3/β-1,6 decasaccharide (Fig 5D and 5E and S7C Fig), suggesting enhanced binding affinity is gained from β-1,6 branching of a β-1,3 backbone polymer. In comparative MST assays, the EPR3 ectodomain binds tested ligands with similar affinities as observed for EPR3a with the exception of a 2-fold lower affinity for *E. bicyclis* laminarin (S10 Fig). β-glucan-induced ROS production has been reported in several plant species [35–37]. We tested whether EPR3a and EPR3 regulate laminarin elicited ROS in *Lotus*. ROS was produced in Gifu by application of laminarin from *L. digitata* and *E. bicyclis* and ROS elicitation was not significantly affected in

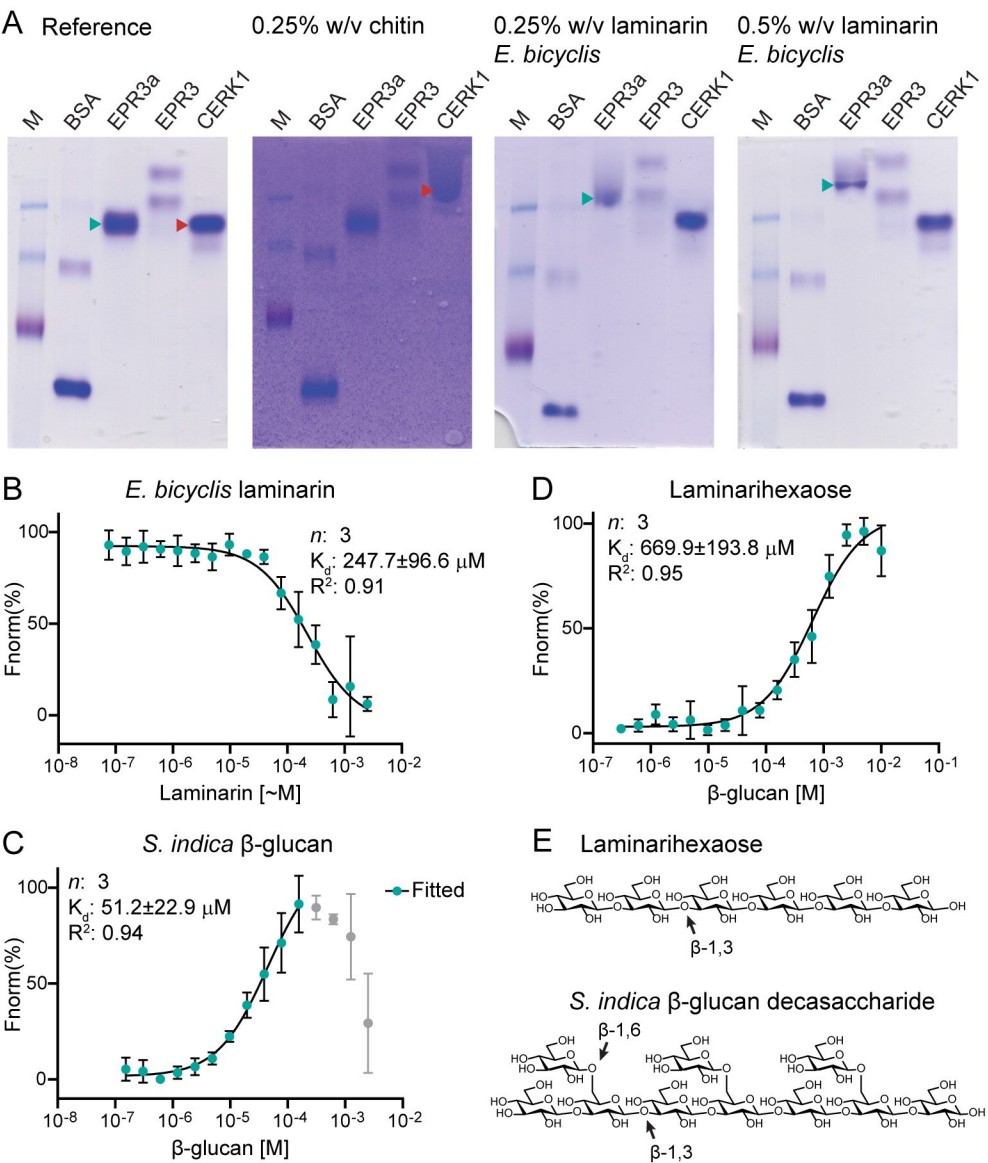

**Fig 5. EPR3a binds fungal-like and fungal-derived β-glucan.** (**A**) An affinity gel electrophoresis assay showed neither EPR3a nor EPR3 ectodomains were retained in gels containing chitin, in contrast to the positive control *At*CERK1 (red arrowheads). The EPR3a ectodomain, but not the EPR3 or the *At*CERK1 ectodomains, was retained in gels containing the β-1,3/β-1,6-glucan laminarin from *E. bicyclis* (green arrowheads). M indicates PageRuler Prestained Protein Ladder, 10 to 180 kDa (Thermo Fisher), and BSA indicates bovine serum albumin, both of which were included as markers to gauge retention of LysM ectodomains. (**B**) Quantitative MST binding analysis showed that EPR3a binds the *E. bicyclis* laminarin with an affinity ≈ 250 μM, which is comparably higher affinity than the laminarin from *L. digitata* (>3 mM, see S7B Fig). (**C**) EPR3a binds a well-defined β-1,3/β-1,6 decasaccharide derived from *S. indica* with an even higher affinity of 51±23 μM. Two binding events were observed, with the high affinity event fitted. (**D**) EPR3a binds the linear β-1,3 laminarihexaose, lacking the β-1,6 branches of the *S. indica* decashaccharide with a low affinity of ≈ 670 μM. (**B–D**) Fnorm(%) is the measured normalised fluorescence of ectodomains assayed over a ligand concentration series, *n* denotes the number of biological replicates, $K_d$ is the calculated dissociation constant, and the goodness of fit is given by $R^2$. (**E**) Chemical structure of laminarihexaose and the *S. indica* β-1,3/β-1,6-glucan decasaccharide [34]. See S1 Data and S1 Raw Images for underlying data. MST, microscale thermophoresis.

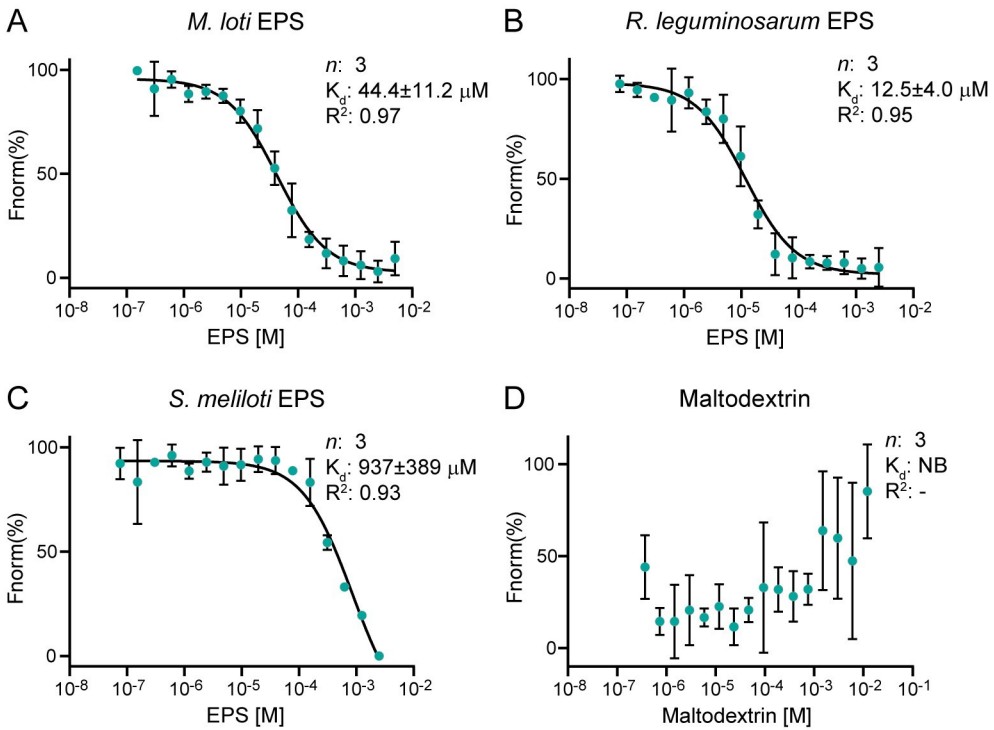

**Fig 6. EPR3a binds EPS from different rhizobia.** MST assays reveal the EPR3a ectodomain binds the repeating EPS octasaccharide unit from symbiotically compatible and incompatible rhizobia. The EPR3a ectodomain bind the EPS from (**A**) *M. loti* and (**B**) *R. leguminosarum* with affinities similar to that of EPR3, as previously reported [22]. EPR3a, however, was observed to bind (**C**) *S. meliloti* EPS with at least 2-fold lower affinity compared to EPR3 [22]. (**D**) No binding was observed for the negative control Maltodextrin. See S1 Data for underlying data. EPS, exopolysaccharide; MST, microscale thermophoresis.

mutant plants (S11 Fig). The *S. indica* β-1,3/β-1,6 decasaccharide did not elicit ROS in Gifu or mutant plants, which is consistent with previous reports [34].

## EPR3a perceives rhizobial exopolysaccharide

The EPR3a receptor is closely related to EPR3, which perceives rhizobial EPS from *M. loti*, *R. leguminosarum*, and *S. meliloti*. Rhizobial EPS, like fungal cell wall glucans, are β-1,6 branched glycans [19,21]. Taking advantage of purified EPS octasaccharides from rhizobia, we investigated if EPR3a ectodomains can also bind rhizobial EPS using quantitative *in vitro* binding assays. MST assays revealed EPR3a binds EPS from *M. loti*, *S. meliloti*, and *R. leguminosarum* with μM affinity, and the affinity for *M. loti* EPS is comparable to that of *S. indica* decasaccharide β-glucan (Fig 6). The observed affinities for rhizobial EPS are comparable to that of EPR3 [22] (S10 Fig) except for a lower affinity for *S. meliloti* EPS. These EPS binding assays indicate that both EPR3a and EPR3 survey EPS present either as a secreted signal or on the surface of rhizobia.

To investigate if EPR3a is also involved in rhizobial EPS perception *in vivo*, symbiotic phenotyping of the *M. loti exoU* EPS mutant strain was carried out. *exoU* produces a truncated form of low molecular weight EPS and is severely impaired in symbiosis with Gifu, being unable to develop fully extended ITs and as a result forming small uninfected nodule primordia [19,20]. *epr3* mutants have been shown to suppress the severe *exoU* phenotype, allowing

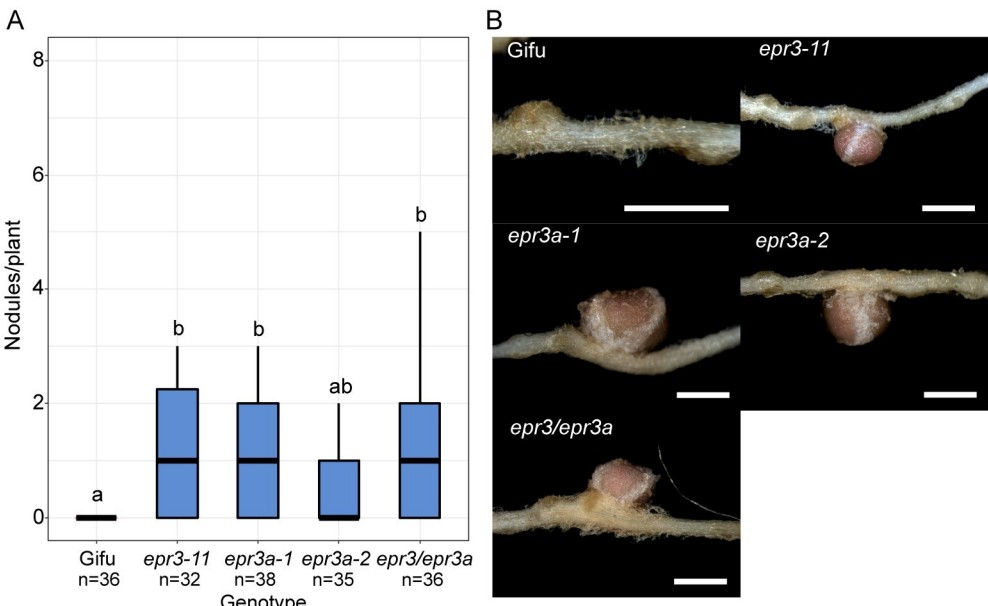

**Fig 7. Nodulation phenotypes following inoculation with *M. loti* R7AexoU. (A)** Nitrogen-fixing nodule formation 5 wpi with *M. loti exoU*. Statistical comparisons between genotypes are shown using ANOVA and Tukey post hoc testing with *p* values (<0.05), as indicated by different letters. **(B)** Representative images of nodules formed on the indicated genotypes following inoculation with *M. loti exoU*. Scale bars are 1 mm. See S1 Data for underlying data.

for the development of mature nitrogen-fixing nodules [19]. Following inoculation with *exoU*, *epr3a* mutants developed mature nitrogen-fixing nodules at a comparable rate to the *epr3-11* mutant (Fig 7A and 7B). No enhancement in the rate of nitrogen-fixing nodule formation is observed in the *epr3a epr3* double mutants compared to the single mutant alleles (Fig 7A). These observed phenotypes following inoculation with *exoU* EPS mutants suggest that EPR3a, like EPR3, is involved in the perception of wild-type and truncated EPS of *M. loti*.

## *epr3a* mutants are impaired in nodule and IT formation

*Epr3a* expression remains at a constitutive low level in *Lotus* root tissues after rhizobial inoculation and during development of nitrogen-fixing root nodules (Fig 8A). In contrast, *Epr3* expression is strongly induced during rhizobial infection of root hairs and cortical tissues/nodule primordia in a Nod factor-dependent manner (Fig 8A) [19,21]. Considering this divergent expression pattern and the similar affinity for rhizobial EPS, we investigated the symbiotic phenotypes of *epr3a*, *epr3*, and *epr3a epr3* double mutants following inoculation with wild-type *M. loti* R7A (R7A).

Quantification of IT formation in the mutants inoculated with R7A revealed a more severe reduction in *epr3a-1* and *epr3a-2* mutants compared to *epr3-11*. *epr3a-1* and *epr3a-2* mutants formed approximately 25% the number of wild-type ITs, while approximately 75% of the wild-type number were observed in *epr3-11* mutants (Fig 8B). Surprisingly, *epr3a epr3* double mutants were less severely impaired than *epr3a* single mutants, forming IT numbers comparable to *epr3-11* (Fig 8B). This result suggests direct interaction between the EPR3a and EPR3 receptors or convergence of downstream signal transduction pathways. We infer that EPR3 promotes IT formation in root hairs and inactivation therefore leads to a reduction of ITs in *epr3* mutants. In the absence of EPR3a, the EPR3 acts negatively, reducing IT formation in

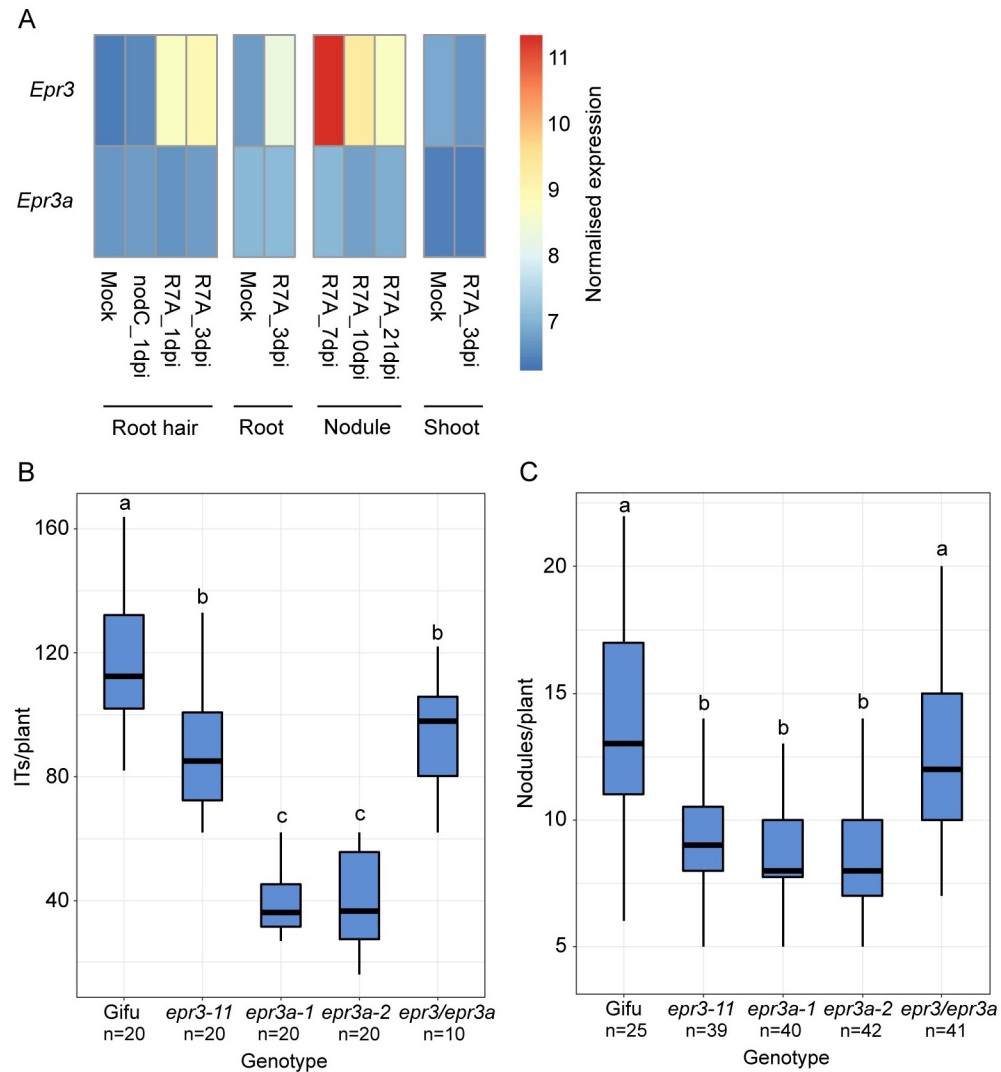

**Fig 8. *Epr3a* expression and symbiotic phenotypes of mutants following R7A inoculation. (A)** Expression of *Epr3* and *Epr3a* in *Lotus* tissues mock-treated or inoculated with *M. loti* R7A. *M. loti nodC* does not produce Nod factor and does not induce symbiosis signalling (control). The normalised expression values presented are from previously obtained RNA-seq data [39,40]. **(B)** IT formation on the indicated genotypes 8 dpi with R7A. **(C)** Nitrogen-fixing nodule formation on the indicated genotypes grown in pots 5 wpi with R7A. Statistical comparisons between genotypes are shown using ANOVA and Tukey post hoc testing with $p$ values ($<0.05$), as indicated by different letters. See S1 Data and S3 Information, NCBI BioProject accession: PRJNA953045 for underlying data. IT, infection thread.

*epr3a* mutants. Inactivation of this negative regulation in double mutants returns IT formation to *epr3* levels.

 *epr3a* mutants show a significant reduction in the number of mature nitrogen-fixing (pink) nodules formed, to a level comparable to the *epr3-11* mutant (Fig 8C). Despite both *epr3a* and *epr3* single mutants exhibiting a significant reduction in nodule formation, an additive effect was not observed in the *epr3a epr3* double mutants. An opposite effect was in fact observed; nodulation was improved in the double mutant compared to single mutants, forming nodules at a level more comparable to wild-type Gifu (Fig 8C). We conclude that autoregulation controlling the number of nodules has been triggered in both *epr3* and *epr3a* mutants and

correlation between ITs and nodule numbers illustrates the high level of synchronisation between rhizobial infection and nodule organogenesis [38].

### EPR3a has an active kinase

LysM receptors are known to harbour either functional catalytically active kinases or pseudo-kinases [41]. In a first step to explore the receptor mechanism, the kinase activities of EPR3a and EPR3 were measured *in vitro*. The intracellular kinases of each were expressed in *E. coli*, purified, and assayed for auto-phosphorylation activity. Kinase assays indicate that both EPR3a and EPR3 contain catalytically active kinases (S12 Fig).

### Working model for EPR3a and EPR3 signalling and interaction in *Lotus*

The current model for LysM receptor function based on chitin receptors and lipochito-oligo-saccharide receptors indicates that signalling is accomplished by receptor complexes, consisting of a receptor with an active kinase and a pseudokinase receptor without kinase activity. Measurable kinase activity of both EPR3a and EPR3 may therefore support a model where they each have a co-receptor rather than working together as a complex. This view is supported by the observed IT formation in the single and double mutants. Inactivation of *Epr3* results in a modest reduction implying that EPR3a can function alone. Inactivation of EPR3a leads to a severe reduction that is reverted in *epr3a epr3* double mutants by inactivation of *Epr3*. This result suggests that EPR3 in the absence of EPR3a is a negative regulator and EPR3a normally acts to counter this negative regulation. Based on our phenotypic characterisation and bio-chemical assays, we cannot determine whether this interaction is the result of a complex forma-tion or if the effect is due to convergence of signalling pathways downstream of the individual receptor complexes (Fig 9). In an attempt to elucidate the downstream signal transduction and gene regulation, we assayed root transcriptional responses in wild-type and mutant plants by RNA-seq. Principal component analysis showed clear separation of samples based on treatment (mock, *M. loti* R7A 3dpi and *M. loti* R7A 7dpi), but no clear separation due to genotype within each treatment cluster (S13A Fig). We analysed the expression of symbiotic genes and found no significant difference between the expression profiles of the wild-type and mutant plants (S13B and S13C Fig). Defence gene expression was investigated by comparing the expression of genes previously identified to be differentially expressed in Gifu in response to the pathogen *Ralstonia* [39]. No clear differences in expression of the defence genes was identified between wild-type and mutant plants (S13D Fig). Our RNA-seq analysis at the whole root level revealed no clear differences in symbiosis and defence-related transcriptional response between wild-type and receptor mutant plants after inoculation with *M. loti* R7A.

## Discussion

Here, we report on the identification and characterisation of a glycan receptor, EPR3a, in *Lotus*. We demonstrate that *Epr3a* expression is induced during AM symbiosis and that *epr3a* mutants show reduced arbuscule formation and increased vesicle formation. This involvement in AM symbiosis clearly separates EPR3a receptor activity from EPR3, which does not appear to have a role in AM symbiosis. LysM receptors previously identified in AM symbiosis include chitin receptors (CERK1 homologues) in rice [42,43], tomato [44] and *Medicago* CERK1 and LYR4 [10,45]. In *Lotus*, no LysM-RLK has previously been identified as involved in AM symbi-osis. The CERK1 homologue CERK6, responsible for immune activation in response to chitin, shows no detectable AM symbiosis phenotype [46]. Expression of *Lys11* was found to be local-ised to cortical cells associated with AM colonisation; however, no reduction in AM symbiosis was observed in *lys11* mutants [47].

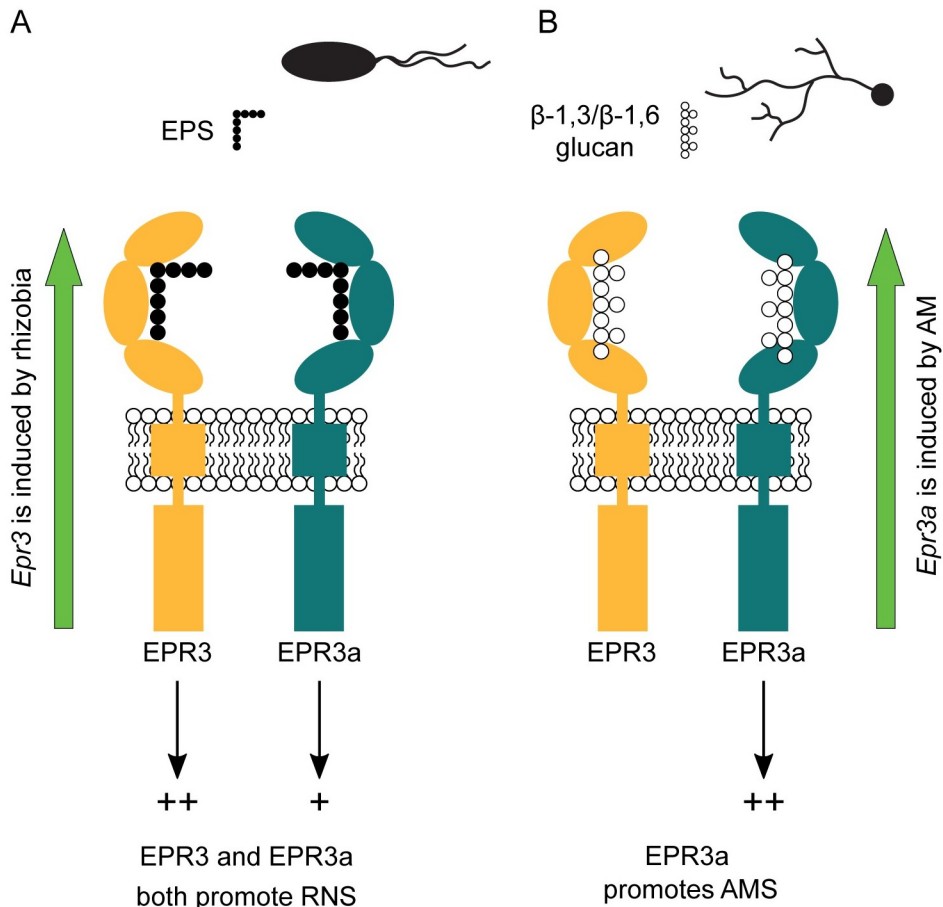

**Fig 9. Working model of EPR3 and EPR3a symbiotic signalling. (A)** In RNS, *Epr3* expression is specifically induced. Both EPR3 and EPR3a can bind rhizobial EPS and promote IT formation and effective nodule development. **(B)** In AMS, *Epr3a* expression is specifically induced. EPR3a alone promotes fungal colonisation, potentially through the binding of a fungal-derived β-1,3/β-1,6-glucan ligand. AMS, arbuscular mycorrhizal symbiosis; EPS, exopolysaccharides; IT, infection thread; RNS, root nodule symbiosis.

As further support for the role of EPR3a in AM symbiosis, we show that the EPR3a ectodomain binds a well-defined β-1,3/β-1,6 decasaccharide derived from the EPS of the endophytic fungus *S. indica* in MST assays and laminarin from *E. bicyclis* with a high degree of β-1,6 branches in affinity gel electrophoresis assays. EPR3 also binds the β-1,3/β-1,6 decasaccharide in MST assays, while binding of *E. bicyclis* glucan in affinity gel electrophoresis assays was not detected for the EPR3 ectodomain. We infer that EPR3 receptor abundance, interaction with downstream components, and glucan binding at the ectodomain are insufficient to affect AM colonisation. To our knowledge, neither EPS nor cell wall glucans from AM fungi have been isolated and purified, limiting our ability to perform quantitative binding assays and determine binding constants. We however show that AM fungi produce β-1,6-glucan in abundance, which is present in the outer cell wall and extracellular matrix surrounding spores and hyphae. Future studies of fungal cell wall components and other fungal glycans are needed to determine the ligand affinities and a possible wider ligand repertoire of EPR3a. The binding of the *S. indica* β-1,3/β-1,6 decasaccharide, the *E. bicyclis* laminarin, and the rhizobial EPS suggests a preference for specific β-1,3/β-1,6 branched glucan structures rather than linear β-1,6 or β-1,3. The low affinity for linear β-1,3-glucan in both MST and gel assays supports this

interpretation. All together, these results suggests that EPR3a high affinity for β-glucans is gained through β-1,6 branches of a β-1,3 backbone.

*Epr3a* expression remains low in root tissues throughout the rhizobial infection process, yet IT and nodulation are impaired in *epr3a* mutant alleles. In contrast, EPR3 is strongly induced by compatible rhizobia, initially in epidermal cells and later within cortical and nodule primordia cells associated with the infecting rhizobia. This expression pattern matches the symbiotic phenotypes of *epr3* mutants, reduced IT formation, and impaired colonisation of nodule primordia [19,21]. However, the reduction in IT formation is more pronounced in *epr3a* than *epr3* mutants. The alleviated symbiotic impairment following inoculation with *M. loti exoU* is comparable in *epr3* and *epr3a* single mutants with no significant enhancement of this in the double mutant. This indicates that although both EPR3 and EPR3a participate in the perception of truncated EPS, they are not exclusively responsible for the symbiotic impairment resulting from the production of incompatible EPS. Our interpretation of these results is that EPR3a, even expressed at low levels, is important for modulating rhizobial infection, potentially through stabilising receptor complex signalling.

Ligand-binding assays support the involvement of EPR3a in monitoring rhizobial EPS during infection, with the EPR3a ectodomain having binding affinities for rhizobial EPS comparable to those reported for the EPR3 ectodomain [22]. The similar binding affinities for EPS ligands isolated from symbiotically compatible *M. loti* R7A and incompatible *S. meliloti* and *R. leguminosarum* suggests that perception of EPS by EPR-type receptors acts in surveillance, rather than specifically recognising compatible symbionts.

Interestingly, the *epr3a epr3* double mutant appears less affected than the respective single mutant alleles in symbiotic events with *M. loti* R7A. This suggests that the downstream signalling from these receptors is altered in the presence/absence of one and other. LysM-RLKs are known to form receptor complexes to mediate signalling activity [48,49]. Whether or not EPR3 and EPR3a function as a receptor complex remains to be resolved. Our analysis of EPR3 and EPR3a intracellular kinase domains reveals that both are catalytic active kinases, which may be of importance when considering the possibility that EPR3 and EPR3a form individual receptor complexes with as-yet unknown co-receptors. Identified LysM-RLK co-receptor pairings generally consist of one catalytically active kinase paired with an inactive kinase, e.g., NFR1 and NFR5 [49,50] or LYK3 and NFP [51].

Within the *Lotus* LysM-RLK family, EPR3a most closely aligns to the EPS receptor EPR3 [19,21]. Structural modelling of EPR3a supports the similarity between the two receptors and, importantly, indicates that EPR3a shares the non-canonical ectodomain structure described for EPR3 [22]. This distinct ectodomain structure presumably explains why these EPR3 and EPR3a LysM-RLKs are able to bind non-GlcNAc containing glycans such as laminarin and rhizobial EPS, instead of traditional GlcNAc containing ligands perceived by LysM-RLKs [46,48,52–57].

Our analysis of EPR-type LysM-RLKs using modelling of the LysM ectodomains identified this distinct receptor type in species across the plant kingdom [22]. The presence of multiple copies of EPR3-type receptors is not restricted to legumes, and some legumes apparently encode a single receptor, e.g., *Medicago*. Given our observation of EPR3a's role in AM symbiosis, a symbiosis that the majority of plants engage in, it is possible that a similar function is performed by homologous receptors in other plant species. An even broader function for EPR3-type receptors in plants surveillance of glycan signal molecules is suggested by binding of the β-1,3/β-1,6 decasaccharide that could be isolated from both the endophytic *Serendipita indica* and pathogenic *Bipolaris sorokiniana* fungi. We propose that the activity of EPR3-type receptors in plants is to monitor surface-exposed glycans produced by microbes during their colonisation, inducing a response different from an immunity response.

## Materials and methods

### Plants and growth conditions

*L. japonicus* ecotype Gifu [58] was used as the wild-type plant. The *epr3-11* mutant has previously been described [19]. *epr3a-1* (30014218) and *epr3a-2* (30155999) LORE1 lines in the *L. japonicus* Gifu accession were obtained through *Lotus* Base [59] with homozygous mutant plants identified as previously described [29,60]. The *epr3/epr3a* double mutant was isolated from crosses between *epr3-11* and *epr3a-2*. Seed sterilisation and plant-growth setups for nodulation and IT assays were as previously described [19]. Plants were grown at 21 ˚C with day and night cycles of 16 and 8 h, respectively. Hairy root transformation was carried out as described previously [61]. Plant growth plate nodulation and IT assays, each containing 10 plants, were inoculated with 750 µl of $OD_{600}$ = 0.02 bacterial suspension. For pot nodulation assays, pots filled with sterile leca and containing 15 plants were inoculated with 25 ml $OD_{600}$ = 0.02 bacterial suspension. For AM studies, a sandwich growth system was used, as previously described [47]. Briefly, seedlings were sandwiched between 2 nitrocellulose membranes (MontaMil Membrane Filters—47 mm, 0.22 µM pores, Frisenette MCE047022) with approximately 100 *Rhizophagus intraradices* spores (Symbiom, CZ) per plant. Sandwiches were planted in sterile quartz (0 to 0.4 mm) in magenta growth containers containing modified Long-Ashton solution (0.75 mM $MgSO_4$, 1 mM $NaNO_3$, 1 mM $K_2SO_4$, 2 mM $CaCl_2$, 3.2 µM $Na_2HPO_4$, 25 µM FeNa-EDTA, 5 µM $MnSO_4$, 0.25 µM $CuSO_4$, 0.5 µM $ZnSO_4$, 25 µM $H_3BO_3$, 0.1 µM $Na_2MoO_4$).

### Bacterial strains and culturing

Wild-type *M. loti* R7A [62,63] and R7AexoU [20] were cultured at 28 ˚C in YMB media. An R7A+pSKDSRED fluorescent reporter strain [64] was used for IT counting. *E. coli* TOP10 was used for cloning and cultured in LB media at 37 ˚C. *Agrobacterium rhizogenes* strain AR1193 [65] was used for hairy root transformation and cultured in LB medium at 28 ˚C.

### Promoter constructs

*Epr3* promoter-reporter constructs have previously been described [19,21]. For *Epr3a*, a 1,979 bp putative promoter region was synthesised based on *L. japonicus* Gifu v1.2 genomic sequence [23]. *Epr3a* promoter, GUS coding sequence, and 35s terminator modules were assembled in pIV10 [66] using GoldenGate cloning [67].

### Expression and purification of EPR3a ectodomain

The C-terminal boundary of the *L. japonicus* EPR3a ectodomain was predicted using the transmembrane helix prediction server TMHMM [68]. The native N-terminal secretion peptide was predicted using the SignalP server [69] and replaced with the *Autographa californica* glycoprotein 67 secretion peptide (MVSAIVLYVLLAAAAHSAFA) [70]. A non-cleavable 6xHis-tag was added to the C-terminal and the EPR3a ectodomain (residues 25–223) fusion construct was codon-optimised for insect cell expression (GenScript, Piscataway, United States of America) and inserted into the transfer vector pOET4 (Oxford Expression Technologies). Recombinant baculovirus was produced in *Sf9* cells using the flashBAC GOLD kit (Oxford Expression Technologies) supplemented with lipofectin (Thermo Fisher) for transfection efficiency. *Sf9* cells were grown to a density of $10^6$ cells/ml in suspension at 26 ˚C in HyClone SFX (GE Healthcare), supplemented with 1% v/v Pen/Strep (10,000 U/ml, Life Technologies) and 1% v/v chemically defined lipid concentrate (Gibco) before infection with a passage 3 baculovirus stock obtaining a MOI = 1–3. EPR3a ectodomain was expressed for 6 days before cell medium

was cleared by centrifugation and dialysed against 50 mM Tris-HCl pH 8.0, 200 mM NaCl at 4 ˚C. EPR3a ectodomain was captured from cell medium utilising a HisTrap Excel column (Cytiva) equilibrated in 50 mM Tris-HCl pH 8.0, 200 mM NaCl. Protein was eluted in 50 mM Tris-HCl pH 8.0, 200 mM NaCl, 500 mM imidazole. EPR3a ectodomain was further purified with a second Ni-AC step using a HisTrap HP column (Cytiva) and purification was finalised with a SEC step using a HiLoad Superdex 75 16/600 pg column (Cytiva) in PBS pH 7.4, 500 mM NaCl. The EPR3 and *At*CERK1 ectodomains were expressed and purified following an identical protocol. EPR3a ectodomain preparation variation was quality controlled by measuring thermostability on a Tycho NT.6 NanoDSF instrument (NanoTemper Technologies) at 1 mg/ml concentration in PBS pH 7.4, 500 mM NaCl. Intrinsic fluorescence was measured at 330 and 350 nm over a temperature gradient (35 to 95 ˚C), and the 330/350 nm ratio was used to produce a transition curve from which an inflection temperature ($T_i$) was calculated. NanoDSF data was analysed with Prism 8 (GraphPad).

## Microscale thermophoresis (MST) binding assays

Heterogenous N-glycosylations of the EPR3a and EPR3 ectodomains negatively affected signal/noise during MST assays, and ectodomain preparations were N-glycan trimmed by PNGase F treatment. PNGase F was incubated with ectodomain in a 1:20 w/w ratio overnight at RT and subsequently removed through SEC. Ectodomain preparations were labelled with Monolith Protein Labeling Kit BLUE-NHS (NanoTemper Technologies) following the manufacturer's instructions. Excess free dye was removed through a Vivaspin 500 centrifugal concentrator (MWCO 10 kDa, Sartorius). A molar labelling ratio of 1–2:1 dye:protein was achieved as estimated by absorbance at 280 and 488 nm using a ND-1000 Nanodrop (Thermo Scientific). MST binding assays were conducted as follows. A 1:1 v/v dilution series of respective ligands was mixed with a final constant concentration of 25 nM labelled protein in PBS pH 7.4, 500 mM NaCl, 0.01% Tween20. Samples were incubated in dark at RT for 30 min to reach steady state. Ligand-ectodomain binding was assayed on a Monolith NT.115 instrument (NanoTemper Technologies) in NT.115 glass capillaries (NanoTemper Technologies) with 20% MST laser power. Data were analysed with MO. Affinity Analysis (NanoTemper Technologies) and normalised data was fitted to a sigmoidal dose-response equation in Prism 8 (Graphpad).

## Affinity gel electrophoresis assay

Approximately 0.1% to 0.5% w/v of carbohydrate polymers were solubilised in separation gel solution (Tris-HCl pH 8.8, 12% acrylamide) by vortexing before polymerisation. Shrimp-shell chitin was homogenised in Tris-HCl pH 8.8 using a pestle head mounted on an electrical stirrer for 5 min before being mixed into the separation gel solution. Affinity gels were cast with a Tris-HCl pH 6.8, 5% acrylamide stacking gel, and 5 μg protein sample loaded in each lane. A Tris/glycine pH 8.3 running buffer was used and gels were run in parallel at 4 ˚C for 5 h with a constant voltage of 150 V.

## Characterisation of laminarins

The *L. digitata* and *E. bicyclis* laminarins were dissolved in 50 mM ammonium acetate buffer and resolved on GE Healthcare Superose 6, 10/300 GL size exclusion column with 50 mM ammonium acetate buffer used as the eluant, at a 0.5 ml/min flow rate. The eluting fractions were monitored with Agilent Technologies 1260 infinity II RI (refractive index) detector. The molecular size was assigned based on the retention time of the polysaccharide standards (1,740 kDa (Vo), 500 kDa, 167 kDa, 67 kDa, 40 kDa, 10 kDa, 5 kDa, and 1 kDa). In addition, the

molecular size of *L. digitata* was confirmed on GE Healthcare Superdex Peptide 10/300 SEC column with 50 mM ammonium acetate used as eluent at 0.5 ml/min flow and with RI detection. The glycosyl linkage analysis of neutral sugars constituting laminarins was determined using the method based on partially methylated alditol acetates (PMAAs) [71]. GC-MS spectra and raw SEC profiles are provided in S4 Information. For the NMR spectroscopy analysis, the *E. bicyclis* laminarin (11.4 mg) and *L. digitata* laminarin (11.3 mg) were dissolved in 90% DMSO (d6-99.96%, Sigma-Aldrich) and 10% $D_2O$ (d-99.9%, Sigma-Aldrich) and heated at 70 ˚C for 5 to 10 min before being transferred to a 5-mm NMR tube (LabScape Stream, Bruker BioSpin AG, Switzerland). No visible aggregates were observed in the samples. The 1D and 2D NMR experiments were recorded as described in the S5 Information.

## Carbohydrate ligands

*M. loti EPS*, *S. meliloti EPS*, and *R. leguminosarum* EPS ligands were obtained as described previously [22,72]. Production and purification of the *S. indica* decasaccharide for binding assay was performed as described in [34]. *L. digitata* laminarin, maltodextrin, and chitin from shrimp shell were purchased from Sigma-Aldrich. *Eisenia bicyclis* laminarin, Pustulan, and Scleroglucan were purchased from Carbosynth. Laminarihexaose, laminaripentaose, and chitohexaose were purchased from Megazyme.

## Protein modelling

The EPR3a ectodomain structure spanning residues 25–223 was modelled using Alphafold2-Colab with a total of 3 recycles [24,25]. An MSA of approximately 1,800 sequences was assembled using MMseqs2 with Uniref and Environmental databases, and no template was used for modelling. Five models were predicted and all models generated a high overall predicted local distance difference test (pLDDT) score of approximately 90. Only the far N- and C-termini showed a pLDDT score <80. The best model had an overall pLDDT score of 90.4 and was used for further analyses. Alphafold models for *Mt*LYK10 (residues 25–221, Uniprot: G7JZ13), *To*EPR (residues 27–221, Uniprot: A0A221I0T5), *Fv*LYK3 (residues 21–211, GenBank: XP_004300916), *Zm*RLK8 (residues 23–232, Uniprot: A0A1D6NMX9), *Hv*RLK9 (residues 24–216, GenBank: KAE8783007), and *At*LYK3 (residues 20–233, Uniprot: F4IB81) were generated similarly as described for EPR3a. Superposition analyses of EPR3-type ectodomains were done using residues spanning the equivalent polypeptide as visualised in the EPR3 crystal structure [22], e.g., residues 28–207 for EPR3a. Structural analyses and figures were prepared using PyMOL Molecular Graphics System, version 2.4 Schrödinger, LCC.

## Histochemical staining and microscopy

IT counts were performed using a Zeiss Axioplan 2 fluorescence microscope using the Zeiss Plan-Neofluar 20x/0.5 objective lens with excitation at 561 nm and an emission filter at 580 to 660 nm. Whole roots were mounted on microscope slides and IT counts performed on a per-root basis. Hairy roots expressing promoter-GUS constructs were GUS-stained as previously described [21]. GUS expression at the whole root level was observed using a Zeiss Discovery V8 stereo microscope. For concomitant visualisation of GUS-promoter activity and AM colonisation, GUS-stained roots were cleared in 10% KOH for 10 min and counterstained with 1 μg/ml WGA-Alexa Fluor 488 (Thermo Fisher, W11261). Bright field and fluorescent images were captured using a Zeiss Axioplan 2 fluorescence microscope with overlaying of the images performed using Fiji ImageJ [73]. For quantification of AM colonisation, roots were stained with 5% black ink (Noir de Jais, Shaeffer) according to [74]. Whole roots were mounted on

microscope slides, and the frequency of arbuscules, fungal hyphae, and vesicles were counted as previously described [75].

## FITC488 labelling and confocal microscopy

Cytological analyses with the confocal laser scanning microscope LEICA SP8 using the chitin-binding WGA and the β-1,6-glucan-binding FGB1 lectins from AM fungi grown in culture with carrot plant material was performed as described in [34], modified from [76].

## Gene expression analysis

Analysis of gene expression by qRT-PCR in *Lotus* during AM symbiosis was performed on a cDNA time series previously generated [47]. qRT-PCR was performed on a LightCycler480 instrument using LightCycler480 SYBR Green I master mix (Roche). ATP and UBC were used as housekeeping reference genes [77]. Quantification of gene levels was calculated using Lin-RegPCR [78]. Three biological replicates were used, each consisting of 5 to 10 plants, and 2 technical replicates were performed. Primer information is given in S2 Table. For RNA-seq, total RNA was isolated from whole roots minus the root tip, using a NuceloSpin RNA Plant kit (Macherey-Nagel) according to the manufacturer's instructions. RNA quality was assessed on an Agilent 2100 Bioanalyser and samples were sent to GATC-Biotech (https://gatc-biotech.com) for library preparation and sequencing on an Illumina MiSeq platform. Mapping was performed using Salmon [79] and gene expression analysed using R-package DeSEQ2 [80], as previously described [81]. RNA-seq raw data has been submitted to NCBI under BioProject accession number PRJNA953045.

## ROS elicitation assays

Gifu, *epr3a-2*, *epr3-11*, and *epr3a-2/epr3-11* seeds were surface sterilised and germinated on water agar for 14 to 16 days and subsequently grown on 1/10 PNM medium for 14 to 28 days. Roots were cut into 0.5 mm pieces and placed into a 96-well plate containing 200 μl 2.5 mM MES per well. Three root pieces were used per well. After recovery overnight, buffer was replaced with 100 μl, 2.5 mM MES containing 20 μM HRP and 20 μM LO12. Following 25 min of incubation, 2× concentrated elicitor solutions were added and chemiluminescence was measured with an integration time of 450 msec using a TECAN SPARK 10M microplate reader, and 2.5 mM MES (mock) and 25 μM chitohexaose (CO6) were used as negative and positive controls, respectively. Laminarin from *Laminaria digitata* and *Eisenia bicyclis* was used at a final concentration of 1 mg/ml and *S. indica* β-glucan decasaccharide was used at a final concentration of 0.5 mg/ml. Total accumulation of ROS was normalised against the averaged mock value, and significant differences were tested using Kruskal–Wallis and post hoc Dunn test.

## Bioinformatics and statistical analysis

*Lotus* LysM-RLK protein sequences were obtained from *Lotus* base [59]. Protein alignment and phylogenetic tree construction were made using CLC Main Workbench 8 (QIAGEN). Protein sequences of EPR-type receptors in diverse plant species were obtained from NCBI. Boxplot generation and statistical analysis were performed in R [82]. Comparison of multiple groups included ANOVA followed by Tukey's post hoc testing to determine statistical significance.

## EPR3a and EPR3 kinase expression, purification, and kinase assays

The EPR3 (residue 259–620) and EPR3a (residue 255–615) intracellular kinase domains were defined as the polypeptide chain C-terminal of the transmembrane helix, as predicted by the TMHMM server [68]. The kinase constructs were expressed in *Escherichia coli* BL21 Rosetta 2 (DE3) cells from a pH10R7Sumo3C vector, containing a 10× histidine tag, a 7× arginine tag, a small ubiquitin-like modifier (Sumo) domain, and a 3C protease cleavage site to allow fusion-tag removal. Cell pellets were resuspended in lysis buffer, 50 mM Tris-HCl pH 8, 500 mM NaCl, 20 mM imidazole, 1 mM benzamidine, 2.5 mM DTT, and 10% (v/v) glycerol. Cell suspensions were lysed by sonication, and the supernatant was collected after pelleting cell debris by centrifugation at 16,000 g, 4 ˚C for 30 min. Supernatants were loaded onto Ni-NTA columns (Macherey-Nagel) and protein was eluted in elution buffer, 50 mM Tris-HCl pH 8, 500 mM NaCl, 500 mM imidazole, 2.5 mM DTT, and 5% (v/v) glycerol. Samples were 3C protease digested and λ-phosphatase de-phosphorylated in dialysis bags overnight at 4 ˚C against dialysis buffer 50 mM Tris-HCl pH 8.0, 200 mM NaCl, 5% glycerol, 1 mM MnCl$_2$, 2.5 mM DTT. Cleaved fusion-tag was removed through a second Ni-AC step and sample purification was finalised by SEC using a Superdex 75 Increase 10/300 (GE Healthcare) column and eluted in 50 mM Tris-HCl pH 8.0, 200 mM NaCl, 2.5 mM DTT. For kinase assays, the dephosphorylated kinases were incubated for 1 h in SEC buffer with and without 10 mM MgCl$_2$ and 20 μM ATP. The samples were loaded onto an SDS-PAGE gel and phosphoproteins were stained using Pro-Q Diamond phosphoprotein gel stain (Invitrogen) according to the manufacturer's protocol. After imaging, the same gel was stained for total protein using SYPRO Ruby Protein Stain (Invitrogen) according to the manufacturer's protocol.

## Supporting information

**S1 Fig. Amino acid alignment of EPR3 and EPR3a.** The "|" indicates perfect alignment, ":" indicates residues of similar properties, "." indicates residues of dissimilar properties, and "-" indicates no alignment.
(TIF)

**S2 Fig. EPR3a contains nonconventional carbohydrate-binding modules as predicted by a high-confidence Alphafold model. (A)** Sequence alignment of the EPR3 and EPR3a M1 domain. Conserved and semi-conserved residues are highlighted in green and light green, respectively. The βαββ secondary structure signature of the EPR3 M1 crystal structure is indicated above the alignment. **(B)** Zoom of the EPR3 M1 crystal structure as compared to the Alphafold generated model of the EPR3a M1. EPR3a shows a strikingly high similarity to the EPR3 M1 with an identical βαββ structure. **(C)** The EPR3a ectodomain Alphafold model shown as a spectrum of the predicted local distance difference test (pLDDT) score. Blue colouring indicates a high pLDDT score and a high confidence in the modelled structure, whereas red colouring indicates a low pLDDT score and low confidence. See S1 Information for underlying data.
(TIF)

**S3 Fig. EPR3-type RLKs are conserved in plants. (A)** One or more EPR3-type RLKs are present throughout plant species. Pseudogenization or lack of gene identification coincides in some cases with the inability of plant species to establish root nodule symbiosis (RNS) or arbuscular mycorrhizal symbiosis (AMS). **(B)** A representative selection of EPR3-type ectodomains throughout plants shows a conserved protein architecture as determined by Alphafold modelling. The EPR3-type M1 (βαββ), M2 (βαβ), and LysM3 (βααβ) are highlighted. Ectodomains are shown in a cartoon representation and are spectrum coloured from blue at the N-

terminus to red at the C-terminus. The RMSD (in Å) of Cα superpositioning to the EPR3 crystal structure is reported and indicates the degree of structural resemblance. See S1 Information for underlying data.
(TIF)

**S4 Fig. *Epr3a* expression in *Lotus* tissues and *Epr3* and *Epr3a* promoter activity in response to AM spore inoculation. (A)** Expression data obtained from the *Lotus* expression atlas (*Lotus* Base http://lotus.au.dk). Epr3a expression is restricted to root tissues with increased expression in response to arbuscular mycorrhiza (AM_27dpi). **(B)** Transgenic roots expressing *pEpr3a*: GUS or *pEpr3*:GUS were inoculated with AM spores. GUS staining was performed on whole root systems 6 wpi. See S1 Data for underlying data.
(TIF)

**S5 Fig. *Epr3a* gene model.** Epr3a gene model with the position of LORE1 insertions in isolated mutant alleles indicated.
(TIF)

**S6 Fig. LysM ectodomain purifications and EPR3a ectodomain NanoDSF quality control.**
**(A)** SEC profile and corresponding SDS-PAGE of the final purification step for the EPR3a ectodomain. Protein elutes as a single peak with an elution volume corresponding to a molecular weight of 32.8 kDa. SDS-PAGE analysis reveals a smeared band between the 25 and 35 kDa marker bands, fitting the weight estimate from SEC and an expected heterogenous N-glycosylated protein preparation. Weight estimates from SEC and SDS-PAGE fit well the theoretical molecular weight of 23.2 kDa for the monomeric protein, plus an additional average of approximately 10 kDa N-glycans. **(B)** A NanoDSF thermal stability assay was used as a quality control for protein preparations and to determine confidence in comparability between replicates in downstream binding assays. The thermal stability of 4 EPR3a ectodomain biological replicates was assayed with NanoDSF, showing that all 4 preparations had similar $T_i$. Each biological preparation was assayed in technical triplicates. **(C, D)** SEC chromatogram and corresponding SDS-PAGE for EPR3 and *At*CERK1 ectodomain purifications. Both preparations, like EPR3a, elute as single peaks fitting monomeric N-glycosylated proteins and migrate as smeared bands between marker bands 25 and 35 kDa in SDS-PAGE. **(A, C, D)** M = molecular weight marker, Inp = input sample of SEC purification. Blue numbering in chromatograms corresponds to different fractions, which are also indicated in the corresponding SDS-PAGE. Blue dashed lines in chromatograms and the horizontal black lines above fraction numbering in SDS-PAGEs indicates the pooled fractions used in biochemical assays. All protein preparations were purified to a high >95% purity as estimated by SDS-PAGE. See S1 Data and S1 Raw Images for underlying data.
(TIF)

**S7 Fig. Supplementary β-glucan affinity gel electrophoresis and EPR3a MST binding data.**
**(A)** Affinity gel electrophoresis assays using β-glucans *L. digitata* laminarin (β-1,3/β-1,6), Scleroglucan (β-1,3/β-1,6), or Pustulan (β-1,6) did not show retention of EPR3a, EPR3, or *At*CERK1 ectodomains. M indicates PageRuler Prestained Protein Ladder, 10 to 180 kDa (Thermo Fisher), and BSA indicates bovine serum albumin, both of which were included as markers to gauge retention of LysM ectodomains. **(B, C)** MST binding data showing EPR3a binds *L. digitata* laminarin and laminaripentaose with low affininty ($K_d$ >3 mM and ≈ 850 μM, respectively). Fnorm(%) is the measured normalised fluorescence of ectodomains assayed over a ligand concentration series, *n* denotes the number of biological replicates, $K_d$ is

the calculated dissociation constant, and the goodness of fit is given by $R^2$. See S1 Data and S1 Raw Images for underlying data.
(TIF)

**S8 Fig. Comparative analysis of the laminarin purified from *L. digitata* and *E.bicyclis*. (A)** Determination of glycosyl linkages in *L. digitata* laminarin (top) and *E.bicyclis* laminarin (bottom). **(B)** The relative distribution (in %) of glycosyl linkages in *L. digitata* and *E.bicyclis* laminarins. **(C)** Determination of the molecular weight of the soluble fraction by size exclusion chromatography on a Superose 6 column. The average MW of *L. digitata* (magenta) is 5,000 Da, and *E. bicyclis* (blue) is approx. 44,000 Da. See S4 Information for underlying data.
(TIF)

**S9 Fig. Residue and linkage composition of *E. bicyclis* and *L. digitata* laminarin with NMR spectroscopy. (A–C)** *E. bicyclis* laminarin contains a 1,3 linked backbone with a complex mix of at least 3 different types of 1,6 branches. **(D–F)** *L. digitata* laminarin is a 1,3 linked main chain with 1,6 branched terminal glucose. **(A)** $^1$H-$^{13}$C-HSQC of *E. bicyclis* laminarin (11.4 mg) in 90% DMSO-d6 and 10% $D_2O$ at 50 ˚C. The signal labelled with a red asterisk belongs to an unknown impurity. **(B)** Based on $^1$H peak integration, the ratio between 1,3 and 1,6 linkages is 3:2. **(C)** Chemical structures found in *E. bicyclis* laminarin. The degree of branching and the spacing of branches cannot be determined. **(D)** $^1$H-$^{13}$C-HSQC of *L. digitata* laminarin (11.3 mg) in 90% DMSO-d6 and 10% $D_2O$ at 60 ˚C. Signals labelled with a red asterisk belong to mannitol located on the reducing end of approximately half of the oligomers, which is consistent with previous studies [32,83]. **(E)** Based on $^1$H peak integration, 1,3:1,6 linkage ratio is 25:1. **(F)** Chemical structure found in *L. digitata* laminarin. The degree and spacing of branching are too ambiguous to determine. See S5 Information for underlying data.
(TIF)

**S10 Fig. EPR3 MST binding data. (A)** The EPR3 ectodomain binds *E. bicyclis* laminarin with atleast 2-fold lower affinity $\approx$ 670 μM compared to EPR3a ($K_d \approx$ 250 μM) when assayed with MST. **(B–G)** EPR3 had similar affinities for other ligands measured as that observed for EPR3a. Binding affinities for **(F)** *M. loti* and **(G)** *R. leguminosarum* EPS was similar as previously reported [22]. **(A–G)** Fnorm(%) is the measured normalised fluorescence of ectodomains assayed over a ligand concentration series, and ΔFnorm(‰) is the normalised difference in fluorescence of experiments with a single biological replicate. *n* denotes the number of biological replicates, $K_d$ is the calculated dissociation constant, and the goodness of fit is given by $R^2$. See S1 Data for underlying data.
(TIF)

**S11 Fig. Laminarin elicited ROS is not affected in *Lotus* mutants *epr3a-2*, *epr3-11*, and *epr3a-2/epr3-11*. (A)** ROS production measured over time in response to mock, chitohexaose (CO6, positive control), and *L. digitata* laminarin. CO6 elicits a fast and strong ROS response in *Lotus* Gifu, *epr3-11* and *epr3a-2*. *L. digitata* laminarin elicits a delayed and relatively weaker ROS response compared to CO6. **(B)** Boxplot of normalised total ROS production measured over 60 min. *L. digitata* laminarin (LD lam) ROS elicitation is not significantly affected in *epr3-11* or *epr3a-2* compared to Gifu. Values represent means ± SEM from 8 wells. Letters represent significant differences based on Kruskal–Wallis and post hoc Dunn test. **(C)** *E. bicyclis* laminarin elicits a relatively faster and weaker ROS burst in Gifu and *epr3-11/epr3a-2* compared to *L. digitata* laminarin. A single well/replicate was performed for the *S. indica* β-glucan decasaccharide in Gifu and *epr3-11/epr3a-2* and no ROS elicitation was detected. **(D)** Boxplot of normalised total ROS production measured over 60 min. *E. bicyclis* (EB lam) and *L. digitata* laminarin (LD lam) ROS elicitation is not significantly affected in *epr3-11/epr3a-2* compared

to Gifu. Values represent means ± SEM from 8 wells. Letters represent significant differences based on Kruskal–Wallis and post hoc Dunn test. See S1 Data for underlying data.
(TIF)

**S12 Fig. EPR3a and EPR3 contain catalytically active kinases. (A)** EPR3a and EPR3 intracellular kinase domains purified from *E. coli* were incubated with or without ATP+MgCl$_2$ and subsequently analysed by SDS-PAGE and Pro-Q Diamond phosphoprotein gel stain. Both EPR3a and EPR3 are able to autophosphorylate, as shown by the enhanced stain intensity in ATP+MgCl$_2$ incubated samples. **(B)** The same SDS-PAGE as in **(A)** stained with SYPRO Ruby total protein stain. BSA (non-phosphorylated) and ovalbumin (phosphorylated) were included as markers for phosphorylation. See S1 Raw Images for underlying data.
(TIF)

**S13 Fig. Gene expression analysis of *epr3* and *epr3a* mutants inoculated with *M. loti* R7A. (A)** Principal component analysis of RNA-seq data obtained from roots of wild-type and mutant plants inoculated with *M. loti* R7A. RNA-seq was performed on root samples harvested 3 and 7 dpi. **(B)** Expression of known symbiotic genes was comparable in wild-type and receptor mutant plants. **(C)** qRT-PCR analysis showed the *NIN* expression profile was comparable in wild-type and receptor mutant plants. **(D)** The 25 *Lotus* genes that show the highest transcriptional response to pathogenic *Ralstonia* were chosen to represent defence-related genes. No significant difference in the expression of the genes was identified between wild-type and mutant plants. See S2 and S3 Information, NCBI BioProject accession: PRJNA953045 for underlying data.
(TIF)

**S1 Table. LORE1 exonic insertions in *epr3a-1* and *epr3a-2* mutant lines.** Only LORE1 inserts in the *Epr3a* gene are shared between the *epr3a-1 and epr3a-2* lines.
(DOCX)

**S2 Table. Primers used in qRT-PCR experiments.**
(DOCX)

**S1 Data. Data.**
(XLSX)

**S1 Information. Alphafold models.**
(ZIP)

**S2 Information. qRT-PCR dataset.**
(ZIP)

**S3 Information. RNA-seq normalised counts.**
(ZIP)

**S4 Information. Laminarin GC-MS and SEC.**
(PDF)

**S5 Information. NMR spectroscopy.**
(DOCX)

**S1 Raw Images. Raw gel images.**
(PDF)

## Acknowledgments

We thank Finn Pedersen for greenhouse assistance and Andreas Prestel for NMR technical assistance. We thank Knud J. Jensen for helpful discussions.

## Author Contributions

**Conceptualization:** Simon Kelly, Simon B. Hansen, Mikkel B. Thygesen, Clive W. Ronson, Alga Zuccaro, Kasper R. Andersen, Simona Radutoiu, Jens Stougaard.

**Data curation:** Simon Kelly, Simon B. Hansen, Mikkel B. Thygesen.

**Formal analysis:** Simon Kelly, Simon B. Hansen, Henriette Rübsam, Kira Gysel, Eva Madland, Artur Muszynski, Parastoo Azadi, Mikkel B. Thygesen, Finn L. Aachmann, Clive W. Ronson, Alga Zuccaro, Kasper R. Andersen, Simona Radutoiu, Jens Stougaard.

**Funding acquisition:** Artur Muszynski, Parastoo Azadi, Mikkel B. Thygesen, Finn L. Aachmann, Clive W. Ronson, Alga Zuccaro, Kasper R. Andersen, Simona Radutoiu, Jens Stougaard.

**Investigation:** Simon Kelly, Simon B. Hansen, Henriette Rübsam, Pia Saake, Emil B. Pedersen, Kira Gysel, Eva Madland, Shunliang Wu, Stephan Wawra, Dugald Reid, John T. Sullivan, Zuzana Blahovska, Maria Vinther, Artur Muszynski, Mikkel B. Thygesen.

**Methodology:** Simon Kelly, Simon B. Hansen, Henriette Rübsam, Pia Saake, Kira Gysel, Eva Madland, Artur Muszynski.

**Project administration:** Simon Kelly, Clive W. Ronson, Alga Zuccaro, Kasper R. Andersen, Simona Radutoiu, Jens Stougaard.

**Resources:** Parastoo Azadi, Finn L. Aachmann, Alga Zuccaro, Kasper R. Andersen, Simona Radutoiu, Jens Stougaard.

**Supervision:** Simon Kelly, Simon B. Hansen, Kira Gysel, Parastoo Azadi, Finn L. Aachmann, Clive W. Ronson, Alga Zuccaro, Kasper R. Andersen, Simona Radutoiu, Jens Stougaard.

**Validation:** Simon Kelly, Simon B. Hansen, Henriette Rübsam, Kasper R. Andersen, Jens Stougaard.

**Visualization:** Simon Kelly, Simon B. Hansen, Henriette Rübsam, Pia Saake, Eva Madland, Artur Muszynski, Jens Stougaard.

**Writing – original draft:** Simon Kelly, Simon B. Hansen, Jens Stougaard.

**Writing – review & editing:** Simon Kelly, Simon B. Hansen, Kira Gysel, Artur Muszynski, Mikkel B. Thygesen, Clive W. Ronson, Alga Zuccaro, Kasper R. Andersen, Simona Radutoiu, Jens Stougaard.

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
