## [Editor Report · Decision Letter 0]

20 Jun 2022

Dear Dr. Stougaard, 

Thank you for submitting your manuscript entitled "A Lotus japonicus glycan receptor kinase facilitates intracellular accommodation of arbuscular mycorrhiza and symbiotic rhizobia." for consideration as a Research Article by PLOS Biology.

Your manuscript has now been evaluated by the PLOS Biology editorial staff, as well as by an academic editor with relevant expertise, and I am writing to let you know that we would like to send your submission out for external peer review.

Once your full submission is complete, your paper will undergo a series of checks in preparation for peer review. After your manuscript has passed the checks it will be sent out for review. To provide the metadata for your submission, please Login to Editorial Manager (https://www.editorialmanager.com/pbiology) within two working days, i.e. by Jun 22 2022 11:59PM.

Kind regards,

Paula

Senior Editor

PLOS Biology

---

## [Decision Letter · Decision Letter 1]

9 Sep 2022

Dear Dr. Stougaard,

Thank you for your patience while your manuscript "A Lotus japonicus glycan receptor kinase facilitates intracellular accommodation of arbuscular mycorrhiza and symbiotic rhizobia." was peer-reviewed at PLOS Biology. Your manuscript has been evaluated by the PLOS Biology editors, an Academic Editor with relevant expertise, and by several independent reviewers.

As you will see in the reviewer reports, which can be found at the end of this email, although the reviewers find the work potentially interesting, they have also raised a substantial number of important concerns. Based on their specific comments and following discussion with the Academic Editor, it is clear that a substantial amount of work would be required to meet the criteria for publication in PLOS Biology. However, given our and the reviewer interest in your study, we would be open to inviting a comprehensive revision of the study that thoroughly addresses all the reviewers' comments. Given the extent of revision that would be needed, we cannot make a decision about publication until we have seen the revised manuscript and your response to the reviewers' comments. Your revised manuscript would need to be seen by the reviewers again, but please note that we would not engage them unless their main concerns have been addressed. 

You will see that the reviewers agree that the manuscript is an interesting article and a potentially important contribution to the field, particularly because there are just a few ECD from RLKs that have been described to bind glycans, but consider that the work would need a substantial revision, additional experiments, and probably a relevant reorientation of the conclusion and models, to provide the strength of advance that PLOS Biology strives to publish. At the end of this letter you will find the comments from the academic editor where the most important points to address are highlighted. 

We appreciate that these requests represent a great deal of extra work, and we are willing to relax our standard revision time to allow you 6 months to revise your study. Please email us (plosbiology@plos.org) if you have any questions or concerns, or envision needing a (short) extension.

**IMPORTANT - SUBMITTING YOUR REVISION**

*Resubmission Checklist*

*Published Peer Review*

*PLOS Data Policy*

*Blot and Gel Data Policy*

Sincerely,

Paula

---

Senior Editor

PLOS Biology

REVIEWS:

Reviewer #1: Rhizobia and Arbuscular Mycorrhizal Symbiosis including receptor signaling

Reviewer #2: Glycan sensing in plants

Reviewer #1: In the manuscript entitled "A Lotus japonicus glycan receptor kinase facilitates intracellular accommodation of arbuscular mycorrhiza and symbiotic rhizobia" Kelly et al. identified a novel LysM-RLK protein EPR3a from L. japonicus. They performed biochemical, molecular, and functional characterization of EPR3a in arbuscular and legume-rhizobia symbioses. In this manuscript, the authors investigated the binding of EPR3a to branched glycans and the perception of rhizobial exopolysaccharide, which is one of the determinants of the specificity of plant-microbe interaction. The authors used LORE1 mutant lines of EPR3a for functional characterization of this protein in arbuscular mycorrhiza association. Finally, the authors used kinase assays to validate in vitro kinase activity of EPR3 and EPR3a. In my opinion, this manuscript suffers from significant flaws. 

1- Many essential controls and critical experiments are missing. This story is highly biased towards symbiosis while it seems likely that EPR3 is a general fungal MAMP receptor:

* The authors performed binding assays between EPR3a ectodomain and glycans obtained from algae (E. bicyclis and L. digitata)... and a fungus irrelevant to the biology of L. japonicus. Since they try to make a case for the role of EPR3a protein in AM symbiosis, the authors should have used fungal extracts from AM fungi and possibly pathogenic fungi relevant to L. japonicus. Using biologically relevant fungal cell wall extracts, the authors would have addressed whether the EPR3a activity is conserved across fungi, involving symbiotic and pathogenic interactions and whether symbiotic or pathogenic interaction dominates in virtue of EPR3a activity.

* The authors checked the nodule number and arbuscular formation, the mature stages of root nodule, and AM symbiosis. Since the protein acts downstream of Nod factor perception, they should have checked the change in expression of genes involved in the SYM pathway, such as NIN and nuclear calcium spiking. Do glycans activate calcium spiking in L. japonicus? Do glycans activate or repress defense signaling in L. japonicus?

* Since the binding seems conserved with algal glycans, it seems likely that glycans from pathogenic fungi will bind EPR3a too. The authors should not frame their story so narrowly around symbiosis and test phenotypes of L. japonicus epr3a mutants with pathogenic fungi too. 

* Since the authors used LORE1 mutants in this study which possess several insertions in the genome, it is not sure if the phenotype obtained is due to non-functional EPR3 and EPR3a only. It is necessary to perform some complementation assays to determine that the observed phenotypes are due to mutations in EPR3a or EPRw, not mutations in other genes.

* The authors performed an RT-qPCR in the root, nodules, and arbuscular mycorrhizal symbiosis. However, they have not shown any data on the expression of EPR3a in other plant tissues or response to fungal pathogens. While missing the expression analysis in other tissues, the authors neglected its role in different plant tissues or biotic interactions.

2- Inconsistencies between the data presented in figures and text. 

* Line 102: "In this study, we report on the identification and characterization of a novel glycan receptor kinase in Lotus that we have designated EPR3a." The authors identified a novel receptor kinase; however, Line 125 says, "A wider search in the genome databases shows that this class of LysM-RLKs is widely conserved in the plant kingdom (22), with either EPR3a or EPR3 or both present in genomes of most plants forming endosymbiosis with mycorrhiza and/or rhizobia (26)." The authors have not presented any data to show the conservation of EPR3a in other plant genomes, and the same is not present in the references cited.

* In the glycans binding assays in Fig. 4, the authors have not used the EPR3 binding ligand. EPR3 binding ligand would have ensured that the EPR3 protein used in the assay was functionally active. Additionally, there are two bands in the EPR3 lanes; can the authors give a plausible reason behind the two bands in the case of EPR3.

* In Supplementary Fig. 6, the authors showed that EPR3 could bind with a low affinity to E. bicyclis glycan; however, the gel in Fig. 4 does not correlate with this statement.

* EPR3a binds with a stronger affinity to S. indica decasaccharide (Kd:~51.2 μM) compared to E. bicyclis glycan (Kd:~247.7 μM), which again indicates that more assays are required to determine the binding affinity of EPR3a with pathogenic or symbiotic signals.

* As per Fig. 5, EPR3a shows higher binding affinity with R. leguminosarum EPS (Kd: ~12.5 μM) compared to M. loti EPS (Kd: ~44.4 μM). This result is contradictory as M. loti is a host for L. japonicus.

3- Some statements are inaccurate: 

* Line 135: "In contrast, Epr3 expression did not differ between mock and AM treatments." however, the expression of EPR3 differed between mock and AM treatments at 2 DPI. 

* In Line 171 authors state, "Arbuscule development in epr3a mutants appears indistinguishable from arbuscule development in wild-type plants, suggesting a role for EPR3a in fungal entry into cortical cells rather than in arbuscule accommodation." however, in line 182 authors make a statement that, "Localisation of Epr3a expression in arbusculated cells and the reduced frequency of arbuscule development in epr3a-1 and epr3a-2 mutants." These two statements are contradictory; maybe the authors wanted to make a statement about arbuscular structure.

Reviewer #2: The article by Kelly et al., describes the identification and characterization of a glycan receptor kinase, EPR3a, closely related to the exopolysaccharide (EPS) receptor EPR3 previously described. EPR3a seems to bind β-1,3 glucans with different β-1,6 branching patterns, which have been described to be characteristic of surface-exposed fungal glucans. EPR3a plays a function in the colonization of Arbuscular Mycorrhiza (AM), since epr3a mutants are defective in different steps of the colonization process. Since cell walls/surface of AM has been suggested to contain β-1,3/β-1,6 glucan, the authors conclude that EPR3a is involved in the perception of this type of glucans from AM. However, purification of these β-1,3/β-1,6 glucans from AM cell walls has not been either performed previously nor in the current article. The demonstration of β-1,3/β-1,6 glucans binding by the extracellular domains (EC), with LysM structures, of EPR2a and EPR has been demonstrated using two approaches: i) affinity gel electrophoresis assays, that is not a very accurate approach, and ii) MST, that is a more precise technology, though clearly has some limitations in EC-glucan binding assays due to precipitation of glucans at high concentration. Based on MST data, the authors conclude that EPR3a binds different types of β-1,3/β-1,6 glucans, but also rhizobial exopolysaccharide (EPS) that is mainly enriched in β-1,6 glucans. Notably, the affinities of EPR3a for the complex mixtures of glucans tested are comparable to those previously described for EPS-EC-EPR3. Though the article lacks the characterization of the specific β-1,3/β-1,6 glucan structure/s recognized by EPR3a and EPR3, the authors show that their ECs bind β-GD, a well-defined β-1,3/β-1,6 glucan present in different fungi. Based on these findings they propose that the activity of EPR3-type receptors in plants is to monitor glycans produced by microbes during their colonization and to regulate intracellular accommodation of microbes and the adaptative response of AM. The results of the article suggest contrasting expression patterns and the authors suggest divergent ligand affinities of EPR3a and EPR3 that explain their distinct functions in AM colonization and rhizobial infection in Lotus japonicus, though these differences in specificity for some ligands tested are quite low and accordingly some conclusion should be weakened. The article contains very interesting new data, and clearly contributes to increase the number of plants RLKs that have been shown to bind glycans, expanding our knowledge in this field of RLK-ligan (glycan) recognition. 

Major Points: 

1. Purified glucans from AM fungal cell walls are not available, therefore, the binding capacity of EPR3a ectodomains was tested in native affinity gel electrophoresis and MST using β-glucans that might or not be present in AM cell walls. The laminarins used in the experiments, that are β-1,3 glucans with β-1,6 branched found in fungal cell wall, are quite complexes in their composition. Though the results of MST seem to be clear, the fact that these very complex mixtures of glucans were used for the binding assays do not contribute to clarify the minimal structure that is perceived by these ECs. Though the article shows that β-GD, a well-defined β-1,3/β-1,6 glucan is bound by both EC receptors, it is not clear if this minimal structure is present in AM cell walls. It would be very interesting to test the binding of short structures (Degree of Polymerizayion of 6 to 12) of pure linear β-1,3 and linear β-1,6 glucan in MST assays or ITC . Some of these pure oligosaccharides are commercially available. These analyses will clarify if β-1,6 branching is really required or not for ECs binding of β-glucans. It is hard to understand at the structural level that ECs of EPR3a and EPR could bind laminarin and EPS.

2. The use of glucan preparations (e.g. scleroglucan, pustulan) enriched in some β-glucans as negative controls might not be appropriate since, as the authors indicate, to work with these polymeric preparations is kind of risky because they contained a diversity of unknown structures and some binding/not binding results could be due to changes in solution viscosity (gelation) and precipitation, which is a well-known property of β-glucans at high concentrations. This is clearly a limitation of working with glucans, but some additional experiments (see 1) to further support authors conclusion could be performed.

3. Interestingly, EPR3a ectodomains can also bind rhizobial EPS from M. loti, S. meliloti, and R. leguminosarum with μM affinity, and the affinity for M. loti EPS is comparable to that of S. indica decasaccharide β-glucan. As indicated in 1, this is quite surprising and interesting, but also indicates that the binding specificity for some glucans is low or the glucan structures present in EPS and laminarin which are bound by ECs are more similar than anticipated. 

4. Surprisingly, epr3a epr3 double mutants were less severely impaired than epr3a single mutants, forming IT numbers comparable to epr3-11, This result suggests direct interaction between the EPR3a and EPR3 receptors or convergence of downstream signal transduction pathway. Based on these results, the authors suggests that EPR3 in the absence of EPR3a is a negative regulator and EPR3a normally acts to counter this negative regulation. However, this hypothesis has not been validated testing the regulation by EPR3a and EPR3 of downstream components of AM and rhizobia colonization pathways. The characterization of downstream events is in general poor in this article. 

Minor Points

1. Legends of Figures with MST data are incomplete. Some concepts like n, different time points, etc. are not well explained in the legend to figures. The values in the Y axe is not explained either.

2. The authors used to ecotypes of Lotus japonica Gifu and Lotus Base (mutants background). Are those genotypes the same? This must be clarified. 

3. Reference 27 is incomplete 

4. Perception of β-GD, a well-defined β-1,3/β-1,6 glucan present in different fungi was found to be perceived by EPR3 and EPR3a receptors, but it has been described to be released by fungi for ROS scavenging beta-glucan to subvert immune responses. How this function of β-GD match with the binding data and functions of EPR3a and EPR3? Is Ros production altered in EPR3a and EPR3 mutants treated with β-GD?

5. The Figure of the expression of EC proteins is not clear explained in Legends to Figures.

COMMENTS FROM THE ACADEMIC EDITOR:

In particular, I would like to see the following major points addressed:

1) the limitations of using of crude preps from algae rather than relevant fungi or purified glucans was brought up by both reviewers and calls into question the specificity of the receptor. This should be addressed with either commercially available purified molecules or crude preps from AM fungi with relevant controls. 

2) I also agree that testing how EPR3a and EPR3 affect downstream regulation of symbiosis signaling is important to put these receptors into the context of the extensive literature. 

3) I do agree that the findings are biased towards AM fungi/symbiosis and it is entirely possible that the proteins also play a role in general immunity against fungi. I think addressing this point from reviewer 1 would strengthen the paper. However, I don’t think their fundings or conclusions preclude this idea and so I think this could also be addressed by modifying the conclusions. 

4) a point-by-point response to all other concerns brought up by reviewers.

---

## [Decision Letter · Decision Letter 2]

30 Mar 2023

Dear Dr. Stougaard,

Thank you for your patience while we considered your revised manuscript "A Lotus japonicus glycan receptor kinase facilitates intracellular accommodation of arbuscular mycorrhiza and symbiotic rhizobia." for publication as a Research Article at PLOS Biology. This revised version of your manuscript has been evaluated by the PLOS Biology editors, the Academic Editor and one of the original reviewers.

Based on the reviews and our Academic Editor's assessment of your revision, we are likely to accept this manuscript for publication, provided you satisfactorily address the remaining points raised by the reviewers. You should respond to the reviewer's concerns and we consider that the issues can be addressed by text modifications. Regarding the issue about "pure structures should be tested”, we think you should clarify earlier in the manuscript that “laminarin, which has the beta-1,6 branched beta-1,3 glucans also found in fungal cell walls” is a mix of different beta-1,6 branched beta 1,3 glucans as you resolve later by NMR. Regarding EPR3 having similar binding affinity as EPR3a (page 27, line 238-240), you show that EPR3a uniquely binds to beta-1,6 branched beta-1,3 glucans while both bind to beta-1,3 glucans with similar affinity, and we think that this does not undermine the central conclusions of the manuscript. Regarding the ROS data, we consider that pathogen assays are beyond the scope of the manuscript.

Please also make sure to address the following data and other policy-related requests.

1. DATA POLICY:

A) Supplementary files (e.g., excel). Please ensure that all data files are uploaded as 'Supporting Information' and are invariably referred to (in the manuscript, figure legends, and the Description field when uploading your files) using the following format verbatim: S1 Data, S2 Data, etc. Multiple panels of a single or even several figures can be included as multiple sheets in one excel file that is saved using exactly the following convention: S1_Data.xlsx (using an underscore).

B) Deposition in a publicly available repository. Please also provide the accession code or a reviewer link so that we may view your data before publication.

Regardless of the method selected, please ensure that you provide the individual numerical values that underlie the summary data displayed in the following figure panels as they are essential for readers to assess your analysis and to reproduce it: Figures 1A, 2A, 3ABC, 5BCD, 6ABCD, 7A, 8ABC, and Supplementary Figures S4A, S6ABCD, S7BC, S8AC, S9BE, S10ABCDEFG, S11ABCD, S13ABCD.

**Please also ensure that figure legends in your manuscript include information on where the underlying data can be found, and ensure your supplemental data file/s has a legend.**

2. We suggest a change in the title to include that Lotus japonicus is a legume: "A glycan receptor kinase facilitates intracellular accommodation of arbuscular mycorrhiza and symbiotic rhizobia in the legume Lotus japonicus".

We expect to receive your revised manuscript within two weeks.

*Published Peer Review History*

*Press*

Sincerely,

Paula

---

Senior Editor,

pjaureguionieva@plos.org,

PLOS Biology

Reviewer remarks:

Reviewer #2: This is a clear improved version of the previous submitted article. Authors have clarified the majority of questions addressed, and have completed the majority of the information that was lacking in the previous article. The articles now shows that EPR3a binds β-1,3 glucans (Lam5 and Lam6) though with lower affinity than β-glucans with β-1,6-branches. However, this analysis is incomplete (additional pure structures should be tested) but based on the β-glucans available for comparison we might accept the experiments performed as those that were feasible. Surprisingly, the difference in specificity of EPR and EPR3a for the decasaccharide β-glucan is very low (it is almost identical).

The RNaseq data clarify some of the questions regarding the transcriptional responses activated. It is a shame that the authors did not include in the RNaseq analyses the beta-glucan that they used for binding experiments for comparison since this had been provided cleaner information about the responses triggered by the ligand and not just by the whole microorganism.

Last, the ROS response observed upon treatment with the β-glucan provides important information. However, since the decasaccharide is not triggering ROS, as described previously, it seems that ROS production observed upon L. digitata laminarin treatment, that is not reduced in EPR and EPR3a mutants, should be triggered by other β-glucan structures distinct from the decasaccharide. The interpretation of the authors might not be correct. I agree with reviewer 2 that the infection of mutants with a Lotus pathogen will contribute to clarify some key aspects of the function of the EPR3a receptor, but these experiments could be the matter of a different history.

---

## [Editor Report · Decision Letter 3]

18 Apr 2023

Dear Dr. Stougaard,

Thank you for the submission of your revised Research Article "A glycan receptor kinase facilitates intracellular accommodation of arbuscular mycorrhiza and symbiotic rhizobia in the legume Lotus japonicus" for publication in PLOS Biology. On behalf of my colleagues and the Academic Editor, Cara Haney, I am pleased to say that we can in principle accept your manuscript for publication, provided you address any remaining formatting and reporting issues. These will be detailed in an email you should receive within 2-3 business days from our colleagues in the journal operations team; no action is required from you until then. Please note that we will not be able to formally accept your manuscript and schedule it for publication until you have completed any requested changes.

PRESS

Sincerely, 

Paula 

---

Senior Editor

PLOS Biology
